# FABLE: Federated Anchor-Based Learning with Privacy Protection

## Abstract

Federated learning enables collaborative model training across distributed clients while preserving their data privacy. However, privacy leakage and data heterogeneity remain significant challenges in federated learning. On the one hand, privacy leakage arises when the exposed information about client models during the client-server communication is exploited to reconstruct sensitive data or misuse client models, compromising both data and model privacy. On the other hand, data heterogeneity limits the generalization capability of the global model on clients, leading to suboptimal performance. Current approaches face a dilemma that stringent privacy constraints degrade the model performance or incur substantial training overhead, while methods addressing data heterogeneity struggle to provide strong privacy guarantees. In this work, to alleviate this dilemma, we propose a novel and simple personalized federated learning method called Federated Anchor-Based LEarning (FABLE), which introduces private anchors during local training. Specifically, clients select private anchors from local datasets to perform an anchor-aware representation transformation, improving the adaptation of the model to local tasks. More importantly, those private anchors not only provide dual privacy protection of data and model privacy, but also avoid significantly computational/communicational overhead or performance sacrifice. Extensive experiments on benchmark datasets under various settings validate the effectiveness of the FABLE method in terms of the privacy protection and model performance.

## 1 Introduction

Federated Learning (FL) (McMahan et al., 2017; Yang et al., 2019; Kairouz et al., 2021) has emerged as a promising distributed machine learning framework, facilitating collaborative model training without compromising client privacy. It accomplishes this by exchanging model information while maintaining data locally. However, the effectiveness of FL is limited by two primary challenges (Li et al., 2020a; Kairouz et al., 2021; Liu et al., 2024a), *i.e.*, privacy leakage and data heterogeneity.

The first challenge in FL, privacy leakage, occurs during the communication between clients and the server. Current studies (Zhu et al., 2019; Zhao et al., 2020) indicate that attackers can reconstruct sensitive data from exposed model information, significantly undermining client data privacy. Additionally, complete exposure of model information during communication can be easily abused without consent (Shokri et al., 2017; Ye et al., 2022; Liu et al., 2023), compromising client model privacy. Another major challenge, data heterogeneity across clients (Zhao et al., 2018; Zhu et al., 2021; Li et al., 2022; Lu et al., 2024; Ye et al., 2024), occurs due to for example varying device conditions and application scenarios. Such data heterogeneity induces substantial variations in locally trained models, hindering global model convergence and resulting in suboptimal performance.

However, under resource constraints, existing studies usually face a dilemma between privacy preservation and data heterogeneity. That is, on the one hand, typical privacy-preserving FL methods (Wei et al., 2020; Truex et al., 2020; Zhang et al., 2020; Byrd & Polychroniadou, 2020; Kanagavelu et al., 2020; Ma et al., 2022) offer stronger privacy guarantees by obfuscating model information during client-server communication. However, they may degrade the model performance or introduce significantly computational/communicational overhead, where data heterogeneity can even worsen such situation. On the other hand, as an important solution for data heterogeneity, personalized federated learning (PFL) customizes personalized models for each client based on their task and device.

Table 1: Comparison of privacy-preserving methods in federated learning across multiple key criteria. The ✓ indicates a clear advantage, while ✗ denotes a clear disadvantage.

| | Data Privacy Protection | Model Privacy Protection | Performance Retention | Data Heterogeneity | Computation & Communication Overhead |
|---|---|---|---|---|---|
| FedAvg | ✗ | ✗ | ✓ | ✗ | ✓ |
| Differential Privacy | ✓ | ✗ | ✗ | ✗ | ✓ |
| Homomorphic Encryption | ✓ | ✓ | ✓ | ✗ | ✗ |
| Secure Multi-Party Computation | ✓ | ✓ | ✗ | ✗ | ✗ |
| FABLE | ✓ | ✓ | ✓ | ✓ | ✓ |

Nevertheless, most PFL methods (Fallah et al., 2020; Dinh et al., 2020; Li et al., 2021b; Wu et al., 2021; Tan et al., 2022a;b; Zhang et al., 2022; Lin et al., 2022; Qu et al., 2023; Ghari & Shen, 2024; Liu et al., 2024b; Li et al., 2025; Zhang et al., 2025; Zheng et al., 2025) provide limited privacy protection, typically ensuring only the data locality.

To kill two birds with one stone, building on model-splitting-based PFL (Arivazhagan et al., 2019; Liang et al., 2020; Collins et al., 2021; Luo et al., 2021; Shang et al., 2022; Xu et al., 2023; Zhang et al., 2023c;a; Yang et al., 2024), we propose a novel and simple FL method called Federated Anchor-Based LEarning (FABLE), which incorporates private anchors during local training. Those anchors offer personalized privacy protection, bridging the gap between robust privacy preservation and enhanced model performance. Specifically, each client selects private anchors from its local dataset without sharing them externally. Then, clients conduct anchor-aware representation transformations using those anchors to perform client-specific coordinate transformations on the shared representation space among client models. This transformation can effectively tailor personalized models to their local tasks, improving the model performance under the data heterogeneity. Additionally, we introduce a linear transformation in the anchor-aware representation transformation, stabilizing the model training and enhancing the model performance.

More importantly, we investigate the effect of private anchors and show that they can provide privacy guarantees at both the data and model levels. Firstly, the introduction of private anchors during local training is equivalent to applying a private anchor-dependent transformation on the original gradients, thereby reducing the risk of recovering sensitive data in gradient inversion attacks. Secondly, we demonstrate that even if attackers obtain the entire model, its performance will be exceptionally poor without private anchors. Thus, private anchors can act as secret keys to protect the model privacy. We compare FABLE with the existing privacy-preserving methods in the Table 1.

We conduct extensive experiments to validate FABLE on various computer vision and natural language datasets under homogeneous and heterogeneous data distributions across clients. The results show that the proposed FABLE method generalizes well across different tasks and achieves comparable performance to competitive baselines. Moreover, empirical evaluations show that the proposed FABLE method could protect both data privacy and model privacy effectively.

In summary, the contributions of this work are three-fold. (1) We propose the FABLE method by introducing private anchors, simultaneously addressing both the privacy leakage and data heterogeneity. (2) The proposed FABLE method provides dual privacy protection through private anchors, safeguarding both data and model privacy without introducing significant computational and communicational overhead or performance degradation. (3) We validate the effectiveness of FABLE through extensive experimentation, demonstrating improvements in model performance across multiple datasets with various settings. Furthermore, we verify that FABLE enhances the privacy protection at both the data and model levels under different attack scenarios.

## 2 RELATED WORK

**Privacy in Federated Learning.** The basic privacy protection mechanism in FL (McMahan et al., 2017) is the model-information-based communication, thereby avoiding the need to share raw data. However, privacy risks arise from the model information exposed during communication, which could be exploited to reconstruct sensitive data (Zhu et al., 2019; Zhao et al., 2020) or leak details about local models (Shokri et al., 2017; Ye et al., 2022; Liu et al., 2023), posing risks to both data and model privacy. Although several advanced privacy-preserving techniques, including differential

privacy (Wei et al., 2020; Truex et al., 2020), homomorphic encryption (Zhang et al., 2020; Ma et al., 2022), and secure multi-party computation (Byrd & Polychroniadou, 2020; Kanagavelu et al., 2020), have been integrated into FL to boost the strength of privacy protection, they typically degrade the performance or impose substantial computation/communication overhead. In this work, we introduce private anchors to provide simultaneous protection for both data and model privacy, thereby preserving both the efficiency and effectiveness.

**Data Heterogeneity in Federated Learning.** Data heterogeneity is a well-known challenge in FL. As an important solution, Personalized Federated Learning (PFL) (Fallah et al., 2020; Dinh et al., 2020; Li et al., 2021b; Wu et al., 2021; Tan et al., 2022a;b; Zhang et al., 2022; Lin et al., 2022; Qu et al., 2023; Xu et al., 2023; Ghari & Shen, 2024; Li et al., 2025; Zhang et al., 2025; Zheng et al., 2025) allows each client to tailor a local model to its own task. Existing PFL algorithms leverage meta-learning (Fallah et al., 2020), regularization (Dinh et al., 2020; Li et al., 2021b; Ghari & Shen, 2024), knowledge distillation (Wu et al., 2021; Tan et al., 2022a;b), and model splitting (Arivazhagan et al., 2019; Liang et al., 2020; Collins et al., 2021; Luo et al., 2021; Shang et al., 2022; Xu et al., 2023; Zhang et al., 2023c;a; Yang et al., 2024; Liu et al., 2024b), and have demonstrated consistent performance gains.

In this work, we focus on the model-splitting-based PFL, which meets both generalization and personalized requirements by dividing models into the public and private components. By exchanging only the public components between clients and the server, this approach significantly reduces communication overhead and increases the difficulty for the attacker to recover private data. However, existing research mainly emphasizes performance optimization under data heterogeneity, with limited exploration of privacy protection mechanisms. To fill this gap, our work amis to improve the model performance while maintaining strong privacy guarantees via private anchors.

**Anchors in Machine Learning.** Anchors act as reference landmarks to establish a latent representation (Norelli et al., 2023; Cannistraci et al., 2023; Crisostomi et al., 2023; Maiorca et al., 2023; Moschella et al., 2023; Cannistraci et al., 2024) and they can be compact yet representative raw data or embeddings. Anchors have emerged as an effective solution of mitigating data heterogeneity in FL (Huang et al., 2024; Zhou et al., 2024; Ye et al., 2023; Dai et al., 2024; Qiu et al., 2024) by stabilizing local training and accelerating global convergence. Unlike those methods that broadcast anchors or their representations, FABLE ensures that private anchors remain exclusively on clients, thereby simultaneously preserving data and model privacy. Moreover, our proposed anchor strategy improves the model accuracy in heterogeneous data distributions while avoiding the performance degradation commonly associated with strong privacy constraints.

## 3 PRELIMINARY

FL enables decentralized clients to collaboratively train a global model under the orchestration of a central server without exchanging raw data. In FL, each client $k \in \{1, \cdots, K\}$ trains a local model $f_k(\theta)$ parameterized with $\theta_k$ on its own dataset $D_k$ consisting of $n_k$ samples by minimizing a local objective function $\mathcal{L}_k(D_k; \theta)$. Then the goal is to obtain a global model $\theta_G^*$ for all clients.

Typical FL preserves privacy by transmitting the model information without exposing raw data during collaborative training. However, existing studies (Zhu et al., 2019) reveal that the sensitive data can be recovered by analyzing model gradients exchanged during the communication. Specifically, attackers leverage the gradient $\mathcal{G}(\theta_k)$ from client $k$ to reconstruct private samples $(x, y) \in \mathcal{D}_k$ by

$$\min_{(x',y')} \left\| \nabla_{\theta_k} \mathcal{L}_k((x', y'); \theta_k) - \mathcal{G}(\theta_k) \right\|^2, \tag{1}$$

where $\mathcal{G}(\theta)$ denotes the operator to obtain the gradient for parameters $\theta$, $\| \cdot \|$ denotes the $\ell_2$ norm of a vector, and $(x', y')$ denotes the reconstruction of one sample and its label. In addition to data leakage, model parameter exposure (Shokri et al., 2017) threatens the intellectual property of each client, *i.e.*, model privacy. That is, during the communication between clients and the server, local model parameters $\theta_k$ are fully visible, increasing susceptibility to unauthorized exploitation by third parties. Consequently, ensuring robust protection of both the data and model privacy in clients remains a critical challenge to the secure deployment of FL.

Another challenge of FL is data heterogeneity, which significantly impacts the optimization and convergence of the collaborative training process. PFL mitigates this issue by tailoring a model

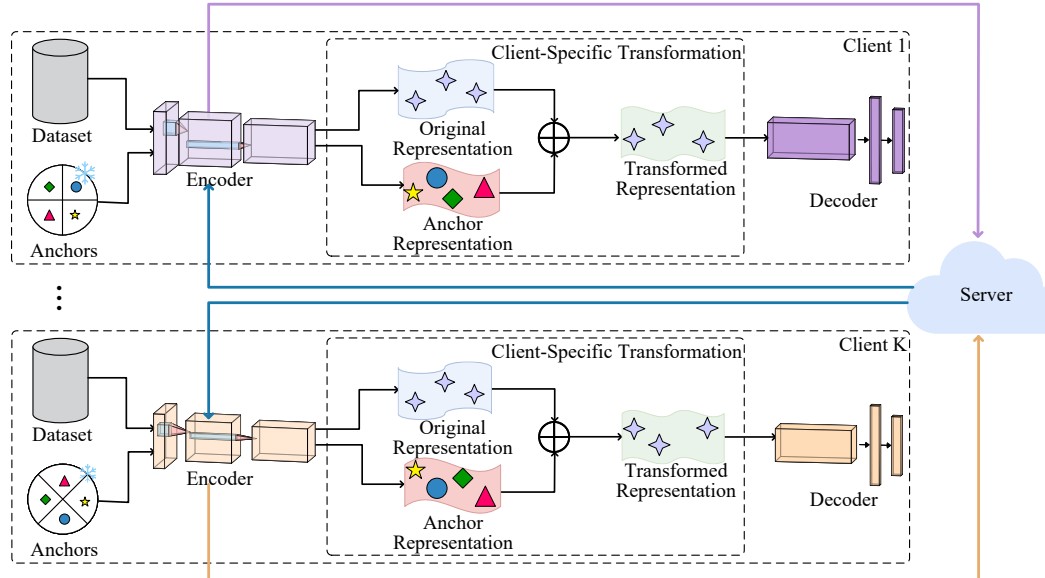

Figure 1: The pipeline of FABLE. Each client maintains a set of fixed personalized anchors which remain unchanged throughout the FL process. Then clients perform client-specific representation transformations by leveraging anchors during local training to transform the original representations into a unique latent space. And clients only exchange the encoders with the server in communication.

$\theta_k^*$ to each client $k$ rather than seeking a single global model $\theta_G^*$ optimal for all clients. In this work, we focus on model-splitting-based PFL, where the model of client $k$ is decomposed into a public encoder $g(\theta_k^g) : \mathbb{X} \mapsto \mathbb{R}^d$, whose parameters could be exposed to the server, and a private decoder $h(\theta_k^h) : \mathbb{R}^d \mapsto \mathbb{Y}$. By limiting the information exposed (*i.e.*, only $\{\theta_k^g\}_{k=1}^K$) during the communication, model-splitting-based PFL not only reduces communication costs but also mitigates the leakage of privacy to a certain extent. Additionally, public encoder capture global knowledge during the collaborative training, whereas private decoders learn personalized knowledge for each client, thereby improving the model performance under both IID and Non-IID settings.

## 4 METHODOLOGY

### 4.1 ANCHOR-AWARE REPRESENTATION TRANSFORMATION

Building on model-splitting-based PFL, we propose Anchor-aware Representation Transformation, which introduces anchors $\mathcal{A}_k$ to perform a client-specific coordinate system transformation $\mathcal{T}_k$ for each client, thus providing a personalized privacy protection by obfuscating the model information.

At initialization, each client independently random selects a set of anchors $\mathcal{A}_k = \{a_i^k\}_{i=1}^{|\mathcal{A}_k|}$ from its local dataset $D_k$. Throughout federated learning process, these anchors remain fixed and undisclosed. After each global updated, each client updates its encoder $g_k(\theta_k^g)$ with the global encoder $g_G(\theta_G^g)$ as the initial starting point, and uses it to encode the private data $x$, while the anchors complete the personalized decoding in combination with local decoder $h(\theta_k^h)$ as

$$
\begin{aligned}
\text{Encoder} &: z^x = g_k(x; \theta_k^g), \\
\text{Decoder} &: \hat{y} = h_k(\mathcal{T}_k(z^x; \mathcal{A}_k); \theta_k^h).
\end{aligned}
\tag{2}
$$

Specifically, we apply a client-specific coordinate system transformation $\mathcal{T}_k$ on $z^x$ to obtain private representation $r_k^x$ by calculating the similarity between $x$ and anchors $\mathcal{A}_k$ as

$$
r_k^x = \mathcal{T}_k(z^x; \mathcal{A}_k) = \left( \text{sim}(g_k(x), g_k(a_1^k)), \dots, \text{sim}(g_k(x), g_k(a_{|\mathcal{A}_k|}^k)) \right),
\tag{3}
$$

where the cosine similarity is used, i.e., $\text{sim}(u, v) = \frac{u^T v}{||u||||v||}$, ensuring angle-based invariance and geometric robustness. We use the transformed representation $r_k^x$ as input to a private decoder $h(\theta_k^h)$ in client $k$, and train the whole model using the cross-entropy loss. During backpropagation, only the data representations $g_k(x)$ are utilized to update the encoder, while anchor representations $g_k(a_i)$ remain static to maintain the training stability. During the inference, clients apply the global encoder with their fixed local anchors to make predictions.

Despite the geometric invariances offered by cosine similarity, the inherent loss of scale information may limit the performance. To address this issue, we propose an additional linear scaling transformation parameterized by a matrix $W_k$ as

$$\hat{r}_k^x = W_k r_k^x. \tag{4}$$

where $W_k$ is locally trained and not externally shared. This transformation corrects scale disparities and enhances personalized performance.

### 4.2 PRIVACY ANALYSIS

**Data privacy.** As a classic attack method in FL, Deep Leakage from Gradient (DLG) (Zhu et al., 2019) can reconstruct sensitive data by using the model gradient of clients in the communication.

In the model-splitting-based PFL, clients only upload the updated gradient of the local encoder $\mathcal{G}(\theta_k^g)$, so the attacker's optimization objective is changed to:

$$\min_{(x', y')} \left\| \nabla_{\theta_k^g} \mathcal{L}_k((x', y'); \theta_k^g) - \mathcal{G}(\theta_k^g) \right\|^2. \tag{5}$$

Model-splitting-based PFL reduces the information exposed compared with basic FL in the communication as shown in Eq. (1), and increases the difficulty for attackers to recover private data.

Built on model-splitting-based PFL, FABLE introduces a client-specific privacy mechanism, which employs a set of private anchors $\mathcal{A}_k$ to perform a personalized coordinate system transformation $\mathcal{T}_k$ on the latent representation $z^x$. During the backpropagation stage, this transformation results in a corresponding private anchor-dependent transformation $\mathcal{F}_k$ being applied to the gradient $\theta_k^g$. Consequently, the optimization objective for an attacker is modified as follows:

$$\min_{(x', y')} \left\| \nabla_{\theta_k^g} \mathcal{L}_k((x', y'); \theta_k^g) - \mathcal{F}_k(\mathcal{G}(\theta_k^g); \mathcal{A}_k) \right\|^2, \tag{6}$$

where $\mathcal{F}_k$ is modulated by the private anchors $\mathcal{A}_k$, which are inaccessible to an adversary. The shared gradient $\mathcal{F}_k(\mathcal{G}(\theta_k^g); \mathcal{A}_k)$ therefore represents an entangled mapping of the private data $x$ and the client-specific anchors $\mathcal{A}_k$. Without knowledge of these anchors, inverting the gradient to recover the original data becomes a severely ill-posed problem. Therefore, FABLE provides a robust privacy enhancement that augments the structural defenses inherent in model-splitting-based PFL.

**Model privacy.** For the sake of simplicity, we discuss model privacy about transmitting the full model. While FABLE will expose less information than FedAvg (McMahan et al., 2017)during communication, a more secure privacy protection is implemented based on the following analysis.

In FABLE, clients control the permission of the model via the private anchors. Therefore, the model prediction is only available when the complete model including anchors is available. Without these anchors, the model's predictions degenerate to the level of random guessing and the attacker cannot obtain authentic model outputs:

$$f_k(x) = \begin{cases} h_k(\mathcal{T}_k(g_k(x); \mathcal{A}_k)), & \text{if } \mathcal{A}_k \text{ is available,} \\ \text{Random Guess}, & \text{if } \mathcal{A}_k \text{ is not available.} \end{cases} \tag{7}$$

The anchor-based privacy protection mechanism significantly enhancing privacy at the model output level, thereby diminishes the effectiveness of membership inference attack (MIA) (Shokri et al., 2017). Consequently, this mechanism in the proposed FABLE method robustly safeguards the model privacy, preserving the intellectual property rights of clients.

Therefore, by introducing anchors, FABLE further provides additional guarantees on both the data and model privacy at a negligible computational and communication cost. For more detailed analysis, please refer to Section A in the appendix.

Table 2: Performance comparison of various methods. The best results are highlighted in **bold**, and the second-best results are underlined.

| | CIFAR 10 | | CIFAR 100 | | Sogou News | | AG News | |
|---|---|---|---|---|---|---|---|---|
| | IID | Non-IID | IID | Non-IID | IID | Non-IID | IID | Non-IID |
| Local | 61.23±0.23 | 92.20±0.10 | 26.27±0.36 | 63.95±0.25 | 93.19±0.04 | 98.24±0.03 | 83.15±0.38 | 96.68±0.02 |
| FedAvg | 83.60±0.34 | 90.36±0.63 | 58.55±0.18 | 50.92±0.22 | 94.19±0.13 | 91.37±0.04 | 91.46±0.15 | 86.10±0.36 |
| FedProx | 85.21±0.08 | 89.32±1.25 | 59.17±0.19 | 56.21±0.39 | 94.32±0.04 | 92.66±0.46 | 91.59±0.16 | 89.85±0.36 |
| SCAFFOLD | 81.91±0.23 | 87.73±0.36 | 56.28±0.36 | 57.41±0.34 | 94.29±0.01 | 93.12±0.15 | 88.75±0.11 | 86.93±0.79 |
| MOON | 83.60±0.21 | 89.76±0.16 | 58.50±0.14 | 50.96±0.57 | 94.27±0.04 | 89.78±0.23 | 91.55±0.14 | 86.88±0.13 |
| FedPer | 82.70±0.30 | 91.46±0.08 | 48.05±1.14 | 72.85±0.20 | 95.07±0.07 | 98.43±0.03 | 90.60±0.18 | 97.45±0.10 |
| Ditto | 83.87±0.14 | 87.96±0.42 | 58.61±0.27 | 71.14±0.46 | 94.26±0.05 | 97.77±0.33 | 91.61±0.12 | 96.65±0.07 |
| FedALA | 83.62±0.31 | 92.94±1.30 | 58.69±0.23 | 67.58±0.29 | 94.91±0.09 | 98.43±0.02 | 91.36±0.09 | 97.64±0.05 |
| GPFL | 79.99±0.65 | 88.99±0.43 | 48.48±0.79 | 70.27±0.45 | 94.95±0.03 | 98.37±0.07 | 91.69±0.07 | 97.67±0.01 |
| FedDBE | 77.27±0.11 | 91.60±0.56 | 49.06±0.30 | 64.34±0.65 | 94.68±0.07 | 98.41±0.01 | 91.93±0.12 | 93.55±0.06 |
| FedAS | 78.82±0.07 | 89.35±1.05 | 54.66±1.19 | 68.17±1.18 | 95.02±0.06 | 98.37±0.03 | 91.79±0.24 | 97.36±0.04 |
| FABLE | 85.45±0.26 | 93.78±0.11 | 57.15±0.04 | 70.73±0.40 | 94.99±0.04 | 98.39±0.02 | 91.42±0.16 | 97.43±0.08 |
| FABLE w/ linear | **85.90±0.10** | **94.34±0.35** | **59.91±0.09** | **73.25±0.06** | **95.26±0.03** | **98.46±0.03** | **91.95±0.19** | **97.95±0.03** |

# 5 EXPERIMENT

## 5.1 EXPERIMENT SETUP

**Datasets and Models.** For computer vision tasks, we investigate image classification on the CIFAR-10 and CIFAR-100 datasets (Krizhevsky et al., 2009) with the ResNet18 and ResNet34 models (He et al., 2016), respectively. For the nature language processing tasks, we use the text classification including the AG News and Sogou News datasets (Zhang et al., 2015), employing a two-layer transformer model (Vaswani et al., 2017) with 512-dimensional embeddings.

**Baselines.** We compare our methods against a variety of competitive baselines for local learning, traditional FL, and personalized FL. Specially, Local learning trains one model per client without any cross-client communication. For traditional FL, we consider FedAvg (McMahan et al., 2017), SCAFFOLD (Karimireddy et al., 2020), MOON (Li et al., 2021a), and FedProx (Li et al., 2020b), which aim to achieve a generalized global model. Additionally, we examine personalized FL methods, such as FedPer (Arivazhagan et al., 2019), Ditto (Li et al., 2021b), FedALA (Zhang et al., 2023c), GPFL (Zhang et al., 2023b), FedDBE (Zhang et al., 2023a), and FedAS (Yang et al., 2024), which customize a model for each client.

**Implementation Details.** In experiments, we use 20 clients and allow each client participating in every communication round. To ensure a fair comparison, the number of anchors in FABLE equals the feature dimension 512 for both vision and language encoders, with the network architecture unchanged. Anchor points for our method are selected via a randomized strategy within each client's local dataset. More details is presented in Appendix B.1.

## 5.2 RESULTS

We report the performance of various FL methods in Table 2. We expanded the experimental setups to include more clients and a wider range of non-IID distributions, and result is shown in Append B.2. Under the IID setting, the Local method consistently yields the lowest performance due to the absence of inter-client collaboration and overfits on the local training dataset. In contrast, traditional FL algorithms exhibit strong performance, benefiting from collaborative training across clients. PFL methods tailor the model to each client, their average performance improvement over FedAvg is marginal and occasionally even inferior. Under Non-IID setting, the Local method becomes more competitive, as it fully adapts to each client's specific task. Meanwhile, traditional FL methods experience notable performance degradation due to client drift caused by heterogeneous data distributions. PFL methods demonstrate increased robustness in this setting.

Vanilla FABLE already outperforms or matches all baselines across most scenarios, regardless of the data heterogeneity, indicating its strong balance between generalization and personalization. Furthermore, the linear variant of FABLE consistently enhances the performance of vanilla FABLE, highlighting the benefit of integrating linear transformations to improve client-specific adaptation.

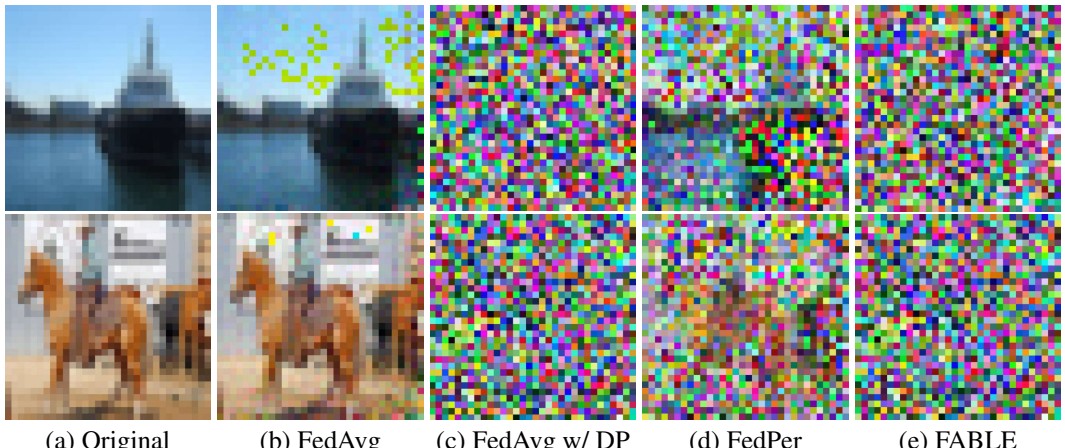

(a) Original     (b) FedAvg     (c) FedAvg w/ DP     (d) FedPer     (e) FABLE

Figure 2: Visualization of reconstructed images using DLG attack in different methods.

## 5.3 ANALYSIS ON PRIVACY PROTECTION

**Data Privacy.** To evaluate the data privacy protection of FABLE, we conduct experiments on the CIFAR-10 dataset using a modified ResNet18 as described in the original DLG method (Zhu et al., 2019). We measure the data privacy protection by Peak Signal-to-Noise Ratio (PSNR) of reconstructed images from DLG attacks. The results are reported in Table 3 and Figure 2.

As shown in Table 3, FedAvg exposes the full model gradients during communication, rendering it highly vulnerable to adversarial reconstruction of private data. Integrating differential privacy with FedAvg mitigates this vulnerability by introducing noise, while comes with a significant drop in model performance. FedPer increases the complexity for adversaries attempting to reconstruct sensitive data by reducing the exposure information during communication. It provides the stronger privacy guarantees compared to FedAvg, while still maintaining competitive model performance. However, FedPer does not alter the gradient information associated with the shared encoder, then adversaries may still approximate the uploaded encoder gradients and partially reconstruct private client data as presented in Figure 2. Based on model-splitting-based PFL, FABLE further advances privacy protection by introducing anchors. The results show that FABLE not only outperforms existing baselines in privacy preservation but also achieves superior accuracy, effectively bridging the gap between privacy and performance in federated learning.

Table 3: Comparison of different methods using modified ResNet18 on CIFAR-10 in terms of model utility and *data privacy* protection. The best results are highlighted in **bold**, and the second-best results are underlined.

Table 4: Comparison of different methods using ResNet18 on CIFAR-10 in terms of model utility and *model privacy* protection. The best results are highlighted in **bold**, and the second-best results are underlined.

|  | Test Accuracy (%) | PSNR |
|---|---|---|
| FedAvg | 74.76±1.25 | 21.04±2.10 |
| FedAvg w/ DP | 59.94±2.29 | **13.83±0.49** |
| FedPer | 73.90±0.52 | 16.37±0.90 |
| FABLE | **80.87±1.13** | 14.81±0.90 |

|  | Test Accuracy (%) | MIA Accuracy (%) |
|---|---|---|
| FedAvg | 83.60±0.34 | 66.12±0.76 |
| FedAvg w/ DP | 77.58±0.09 | 59.65±0.87 |
| FedPer | 82.70±0.30 | 56.60± 0.90 |
| FABLE | **85.45±0.26** | **53.97±0.66** |

**Model privacy.** In this section, we evaluate the impact of anchor availability on model performance during inference, with the result presented in Figure 3. We observe that even with complete access to model parameters, the absence of anchors during the inference phase results in a significant performance drop, approaching random guessing in some scenarios. Therefore, we propose leveraging anchors as secret keys, preserving model performance and improves model privacy by preventing unauthorized inference during deployment.

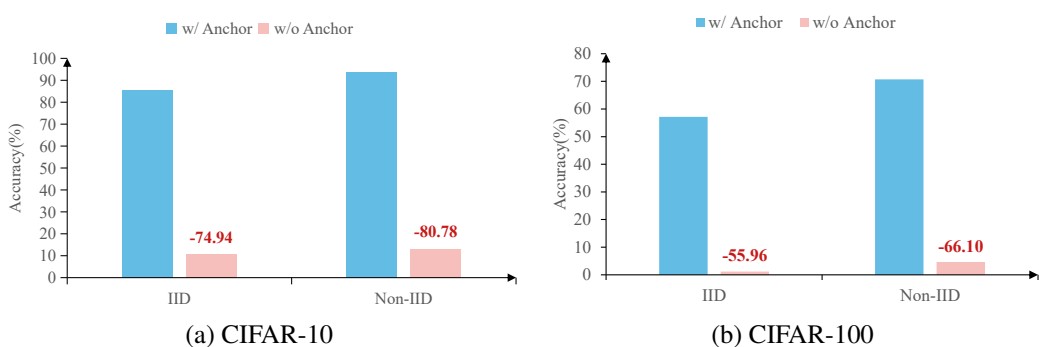

Figure 3: Impact of anchor availability on model inference accuracy evaluated.

To further evaluate the privacy protection strength of FABLE, we conduct Membership Inference Attacks (MIA) (Shokri et al., 2017) on CIFAR-10 using ResNet18, and show the result in Table 4. As illustrated in Table 4, FedAvg exposes the entire model, enabling adversaries to train effective shadow models for membership inference. In contrast, FedAvg combined with differential privacy introduces noise to the model outputs, mitigating attack success but at the expense of notable performance degradation. FedPer, which only transmits encoder during communication, reduces the attacker's confidence by limiting access to sensitive model representations. FABLE further enhances privacy protection by ensuring clients do not share anchor. Even with full model access, adversaries are restricted to unreliable outputs, significantly reducing the accuracy of MIA. These results demonstrate that FABLE effectively achieves model-level privacy protection without compromising utility. Furthermore, we conduct additional experiments to explore the impact of different anchor counts on the trade-off between privacy and model accuracy in Append B.4.

## 5.4 ABLATION STUDY

We conduct an ablation study to investigate the impact of the number of anchors on the model performance. Additionally, we introduce a linear transformation to other baselines and evaluate its influence alongside our proposed approach. Furthermore, we compare the impact of different anchor selection strategies. We show the results in Tables 5, 6 and 7.

Table 5: Evaluation of model performance with varying numbers of anchors. The best results are highlighted in **bold**, and the second-best results are underlined.

| Anchors Numbers | | 128 | | 256 | | 512 | | 1024 | |
|---|---|---|---|---|---|---|---|---|---|
| | | IID | Non-IID | IID | Non-IID | IID | Non-IID | IID | Non-IID |
| CIFAR-10 | FABLE | 85.29±0.15 | 92.78±0.06 | 85.38±0.06 | 93.32±0.08 | 85.45±0.26 | 93.78±0.11 | 85.12±0.26 | 93.78±0.09 |
| | FABLE w/ linear | 85.73±0.08 | 93.47±0.16 | 85.80±0.10 | 93.64±0.19 | 85.90±0.10 | 94.34±0.35 | **85.96±0.06** | **94.36±0.04** |
| CIFAR-100 | FABLE | 56.49±0.28 | 67.82±0.24 | 56.89±0.25 | 69.27±0.45 | 57.15±0.04 | 70.73±0.40 | 56.70±0.23 | 71.98±0.35 |
| | FABLE w/ linear | 59.24±0.18 | 70.43±0.23 | 59.61±0.09 | 72.46±0.20 | 59.91±0.09 | 73.25±0.06 | **59.96±0.17** | **73.63±0.34** |

**The Effect of Anchors Number.** As illustrated in Table 5, the overall performance of the model improves as the anchors number increases. However, when the anchors number increases from 512 to 1024, the performance gains are minimal and even slightly degraded in some cases. The dimensionality of the transformed representation space is directly correlated with the number of anchors. When the anchors number increases, enabling the model to capture richer representations in the higher dimensional space. Nevertheless, when anchors number surpasses a certain threshold, which introduces noise in the high-dimensional space, affecting the model's discriminative capability.

**The Effect of Linear Transformation.** In Table 5, we also observe that adding a linear transformation layer stabilizes the model's performance compared to the original transformation, particularly when the number of anchors is large. This mapping acts as a dimensionality optimization step, mitigating noise interference. As a result, the model not only maintains stable performance but also benefits from the enhanced expressiveness of the transformed representation space.

Table 6: Performance comparison of different methods with and without linear transformation. The values in parentheses represent the performance change introduced by the linear transformation.

| | CIFAR-10 | | CIFAR-100 | |
|---|---|---|---|---|
| | IID | Non-IID | IID | Non-IID |
| FedAvg | 83.60±0.34 | 90.36±0.63 | 58.55±0.18 | 50.92±0.22 |
| w/ linear | 74.70±0.73(-8.90 ↓) | 90.88±0.09(+0.52 ↑) | 49.52±0.26(-9.03 ↓) | 55.65±0.06(+4.73 ↑) |
| FedPer | 82.70±0.30 | 91.46±0.08 | 48.05±1.14 | 72.85±0.20 |
| w/ linear | 74.98±1.09(-7.72 ↓) | 91.90±0.09(+0.44 ↑) | 47.55±0.15(-0.50 ↓) | 73.03±0.16(+0.18 ↑) |
| FABLE | 85.45±0.26 | 93.78±0.11 | 57.15±0.04 | 70.73±0.40 |
| w/ linear | 85.90±0.10(+0.45 ↑) | 94.34±0.35(+0.56 ↑) | 59.91±0.09(+2.76 ↑) | 73.25±0.06 (+2.52 ↑) |

Additionally, as shown in Table 6, other baselines benefit from linear transformation in Non-IID setting, but experience performance degradation under IID settings, possibly due to the distortion introduced by added transformation layers. FABLE effectively integrates the linear transformation with its anchor-aware representation, maintaining robustness under IID and Non-IID settings. The results suggest that the linear transformation in FABLE not only enriches feature representation but also mitigates the scale mismatch introduced by anchor-aware representation transformation.

Table 7: Performance comparison of different anchor selection strategies. The best results are highlighted in **bold**, and the second-best results are underlined.

| | | CIFAR-10 | | CIFAR-100 | |
|---|---|---|---|---|---|
| | | IID | Non-IID | IID | Non-IID |
| fps | FABLE | 85.31±0.09 | 93.82±0.12 | 58.54±0.12 | 70.84±0.10 |
| | FABLE w/ linear | **85.92±0.02** | 94.43±0.11 | 59.67±0.32 | **73.26±0.17** |
| $k$-means | FABLE | 85.15±0.21 | 93.82±0.29 | 58.78±0.49 | 70.89±0.26 |
| | FABLE w/ linear | 85.85±0.27 | **94.54±0.26** | 59.90±0.35 | 73.14±0.02 |
| random | FABLE | 85.45±0.26 | 93.78±0.11 | 57.15±0.04 | 70.73±0.40 |
| | FABLE w/ linear | 85.90±0.10 | 94.34±0.35 | **59.91±0.09** | 73.25±0.06 |

**The Effect of Anchor Selection Strategy.** Illustrated in the Table 7, in addition to random selection, we introduce two new strategies for anchor selection: farthest point sampling (fps), where we select anchors based on the farthest points from each other, and $k$-means clustering, where we select anchors closest to the centroids of $k$ clusters and $k$ is equal to the number of anchors. We can see that the performance of different anchor selection strategies is quite comparable. We believe that random sampling is the most practical strategy as it does not require additional computation for the selection process and still yields competitive performance.

## 6 CONCLUSION

In this paper, we propose Federated Anchor-Based LEarning (FABLE) to addresses the dual challenges of privacy leakage and data heterogeneity in Federated Learning. By leveraging anchors as secret keys, FABLE performs client-specific coordinate transformations on the original representation spaces, which provides personalized protection for data and models while enhancing model adaptability to local tasks. To mitigate the impact of scale normalization introduced during representation transformation, we further incorporate an additional linear transformation, which stabilizes model training and improves overall performance. Experimental results demonstrate that FABLE maintains competitive model accuracy and offers dual privacy guarantees for both data and model privacy. We believe FABLE offers a new perspective on privacy-preserving personalized federated learning that can be extended to real-world applications.

ETHICS STATEMENT

This paper presents work whose goal is to advance the field of Machine Learning by improving the privacy protection of AI systems. The authors have read and comply with the ICLR Code of Ethics. The research did not involve human subjects, animal experiments, or personally identifiable data. All experiments were conducted on publicly available benchmarks and open-source models. We have carefully considered the broader impacts and believe that this work poses no foreseeable risks of harm while contributing to the development of privacy-preserving machine learning.

REPRODUCIBILITY STATEMENT

To ensure the reproducibility of our research, we provide a comprehensive description of our methodology and experimental setup. The proposed Federated Anchor-Based LEarning (FABLE) algorithm, including the anchor-aware representation transformation and the optional linear scaling, is detailed in Section 4. Our full experimental setup, including the datasets, model architectures, and data partitioning strategies for both IID and Non-IID settings, is described in Section 5.1 and expanded upon in Appendix B.1. Crucial implementation details, such as the number of clients, communication rounds, local training epochs, batch sizes, and all hyperparameters for our method and the baselines, are also provided in Appendix B.1. During the reviewing process, the source code is supplied anonymously as part of the supplementary materials. Additionally, upon the acceptance of the paper, this code will be publicly released.

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

## A  PRIVACY ANALYSIS

**Data Privacy.**    To understand the strength of data privacy protection of different federated learning (FL) methods under DLG attack (Zhu et al., 2019), we analyze the corresponding gradient information in the communication process of various methods.

In the standard FL scenario (McMahan et al., 2017), each client $k$ computes the gradient based on its local dataset and communicates the complete gradient $\mathcal{G}(\theta_k)$:

$$\mathcal{G}(\theta_k) = \nabla_{\theta_k} \mathcal{L}_k(h_k(g_k(x)); y) \tag{8}$$

This gradient fully captures the local data characteristics, making it vulnerable to DLG attacks, where attackers optimize to match gradients directly and reconstruct the original data.

In model-splitting-based personalized federated learning (PFL), clients partition the model into a public encoder $g_k$ parameterized by $\theta_k^g$ and a private encoder $h_k$ parameterized by $\theta_k^h$, where $\theta_k = [\theta_k^g; \theta_k^h]$. Clients only communicate the gradient of the public encoder:

$$\begin{aligned} \mathcal{G}(\theta_k^g) &= \nabla_{\theta_k^g} \mathcal{L}_k(h_k(g_k(x)); y) \\ &= \frac{\partial \mathcal{L}_k}{\partial g_k(x)} \cdot \frac{\partial g_k(x)}{\partial \theta_k^g}. \end{aligned} \tag{9}$$

Unlike standard FL, this approach only partially exposes gradient information, thus theoretically limiting the efficacy of DLG attacks.

In FABLE, the anchor representation remains fixed during local training, and gradient updates to the public encoder occur without backpropagation through anchor representations. Consequently, this anchor-aware transformation introduces additional obfuscation into the gradient information, which is equivalent to a client-specific linear mapping $\mathcal{F}_k$:

$$\begin{aligned} \mathcal{G}_{\text{FABLE}}(\theta_k^g) &= \nabla_{\theta_k^g} \mathcal{L}_k(h_k(\mathcal{T}_k(g_k(x); \mathcal{A}_k); y) \\ &= \underbrace{\frac{\partial \mathcal{L}_k}{\partial r_k^x} \frac{\partial \mathcal{T}_k(g_k(x), g_k(\mathcal{A}_k))}{\partial g_k(x)}}_{\mathcal{F}k} \frac{\partial g_k(x)}{\partial \theta_k^g} \\ &= \mathcal{F}_k(\mathcal{G}(\theta_k^g); \mathcal{A}_k) \end{aligned} \tag{10}$$

This formulation effectively conceals the original encoder gradient $\mathcal{G}(\theta_k^g)$. In FABLE, the gradient visible to an attacker $\mathcal{G}_{\text{FABLE}}(\theta_k^g)$ is not the original gradient but a transformed version, $\mathcal{F}_k(\mathcal{G}(\theta_k^g); \mathcal{A}_k)$. This transformation is dependent on the private anchors $\mathcal{A}_k$ which are known only to the client. As a result, the search space for a potential attack is drastically expanded. An adversary must now attempt to simultaneously infer not only the private data sample $(x', y')$ but also the entire set of anchor data points $\mathcal{A}_k$. Given that the anchors are randomly selected from the client's local dataset, the space of potential combinations is combinatorially vast, rendering a brute-force or gradient-based search for $\mathcal{A}_k$ computationally infeasible. Consequently, any single observed gradient $\mathcal{G}_{\text{FABLE}}(\theta_k^g)$ could correspond to a multitude of different data and anchor pairs $(x, \mathcal{A}_k)$, which confounds the attacker's optimization process. Therefore, FABLE based on the inherent structural defense of model-splitting based PFL, provides powerful privacy protection measures.

**Model Privacy.**    Membership inference attacks (MIA) (Shokri et al., 2017) aims to determine whether a given data sample is part of a client's private training dataset. Specifically, given an input $(x, y)$, the adversary estimates the probability that this sample belongs to the client dataset $\mathcal{D}_k$ using:

$$P\left((x, y) \in \mathcal{D}_k\right) = f_{\text{attack}}((x, y)). \tag{11}$$

The attacker model $f_{\text{attack}}$ is trained on the dataset $\mathcal{X}$, which is constructed based on the outputs of the target model to distinguish between member and non-member samples. This process relies heavily on the availability of accurate model predictions, highlighting the importance of safeguarding the informativeness of these outputs.

FABLE introduces a cryptographic-style mechanism for output regulation, while only clients $k$ possess the private anchor $\mathcal{A}_k$ can compute valid predictions using the complete model $f_k$. In the

Figure 4: Comparative of privacy protection in classical FL and FABLE. The above panel shows that attackers can exploit model to reconstruct private data, and misuse local models without permission in classical FL. The below panel presents FABLE integrates anchors as secret keys to perform client-specific feature transformation. It not only renders gradient inversion attacks ineffective but also prevents unauthorized model exploitation, ensuring robust protection of both data and model privacy.

absence of $\mathcal{A}_k$, the model's output $f_k(x)$ degenerates into randomized responses. As the attack model's training data $\mathcal{X}$ is derived from these outputs, the absence of $\mathcal{A}_k$ ensures that the predictions for member and non-member samples become indistinguishable. This injects significant noise into $\mathcal{X}$, severely degrading the performance of the attack model $f_{attack}$ and collapsing the separability required for successful membership inference.

This anchor-conditioned inference pipeline introduces a privacy-preserving inductive bias. It means that only authorized prediction pathways yield meaningful outputs, while unauthorized ones result in randomized behavior despite full parameter access. As a result, FABLE offers robust protection against inference attacks by mitigating output-level leakage. Furthermore, it safeguards the intellectual property of deployed models by rendering unauthorized usage functionally ineffective.

In summary, FABLE enhances both data-level and model-level privacy without compromising utility for legitimate clients, who retain exclusive access to their private anchors $\mathcal{A}_k$. The comparative advantages of FABLE over traditional FL in terms of privacy protection are illustrated in Figure 4.

# B  EXPERIMENT

## B.1  EXPERIMENT SETUP.

**Datasets and Models.**   We evaluate our method on a diverse set of datasets spanning both image and text classification tasks:

- CIFAR-10: The dataset comprises 60,000 images of size 32×32 pixels, evenly distributed across 10 distinct classes. It is divided into 50,000 training and 10,000 test images.

- CIFAR-100: An extension of CIFAR-10, CIFAR-100 also includes 60,000 32×32 images, but spans 100 fine-grained categories. The data is split into 50,000 training and 10,000 test

samples. Each image is annotated with both fine and coarse labels; in our experiments, we utilize the fine labels for model training.

- AG News: A benchmark dataset for topic classification, AG News contains 120,000 training and 7,600 testing samples, each comprising a news headline and a brief description. The articles are categorized into four topics.

- Sogou News: This is a large-scale Chinese news classification dataset, including 90,000 training and 12,000 test samples, labeled across five categories. Each sample consists of a Mandarin-language headline and body text.

These datasets enable a comprehensive evaluation of our method under both IID and Non-IID settings, allowing us to assess the model's generalization and adaptability across multiple modalities.

We simulate the FL environment under both IID and Non-IID settings. In the IID setting, each client receives an equal-sized, class-balanced shard sampled uniformly from the entire dataset. In the Non-IID setting, we sample data from datasets using the Dirichlet distribution (Lin et al., 2020), denoted as $Dir(\beta)$. Specifically, we sample $q_{c,i} \sim Dir(\beta)$ and allocate $q_{c,i}$ proportion of samples with label $c$ to client $i$, where $\beta = 0.1$ for all tasks. To simulate a typical FL scenario, where data is only available on clients, we randomly split the local data on each client into 80% for training and 20% for testing, and report all results on the test data.

**Baselines.** To comprehensively assess the performance of our approach, we compare it against several representative federated learning (FL) and personalized federated learning (PFL) baselines:

- FedAvg (McMahan et al., 2017) : Averages client-side stochastic gradient updates to construct a global model.

- SCAFFOLD (Karimireddy et al., 2020): Utilizes control variates at both server and client levels to correct for client-drift caused by data heterogeneity, leading to more stable and faster convergence.

- MOON (Li et al., 2021a): Leverages model-level contrastive learning to correct local training by maximizing the agreement between representations from the local model and the global model, mitigating drift from non-IID data.

- FedProx (Li et al., 2020b): Enhances FedAvg by introducing a proximal term to address data and system heterogeneity, improving convergence stability.

- FedPer (Arivazhagan et al., 2019): Employs model splitting by sharing a global encoder while maintaining personalized decoders on each client, thereby preserving client-specific characteristics.

- Ditto (Li et al., 2021b): A multi-task learning framework that learns both a personalized and a global model for each client, using a regularization term to enforce consistency between them, thereby inherently improving fairness and robustness.

- FedALA (Zhang et al., 2023c): Adapts model aggregation via learned element-wise mixing weights, effectively capturing client preferences by interpolating between local and global models.

- GPFL (Zhang et al., 2023b): Simultaneously learns global and personalized features on each client by introducing a Conditional Valve (CoV) to create distinct routes for global and personalized tasks, guided by shared Global Category Embeddings (GCE).

- FedDBE: Disentangles personalized biases from global knowledge using a memory module and regularization term, promoting robust bi-directional knowledge transfer.

- FedAS (Yang et al., 2024): Addresses inconsistencies by using parameter-alignment to infuse local knowledge into global parameters and client-synchronization, which leverages the Fisher Information Matrix to down-weight contributions from under-trained clients.

For all methods, we report test accuracy as the primary evaluation metric. In local training and PFL settings, we report the accuracy of the best-performing model across training epochs. For traditional FL settings, we report the accuracy of the best single global model. Each experiment is repeated three times using different random seeds, and we report the mean and standard deviation.

**Implement Details.** The number of communication rounds is set to 500, and the local training epoch is set to 1. During local training, we use a batch size of 64 for computer vision datasets and 256 for natural language processing tasks. The models are optimized using SGD with a momentum of 0.9 and weight decay of 5e-4. We perform grid search over learning rates in the range [0.01, 1] to select the optimal configuration for each baseline. A cosine learning rate scheduler is applied throughout training. All experiments are implemented using PyTorch 2.0, based on PFLlib (Zhang et al., 2023d), and are conducted on an NVIDIA 3090 GPU.

Table 8: Performance comparison of different methods with and without linear transformation under diverse settings. The best results are highlighted in **bold**, and the second-best results are underlined.

| | CIFAR-10 | | | CIFAR-100 | | |
|---|---|---|---|---|---|---|
| | IID | Dir(0.5) | Dir(0.01) | IID | Dir(0.5) | Dir(0.01) |
| FedAvg | 82.19±0.19 | 71.53±1.00 | 70.83±0.49 | 51.71±0.17 | 51.17±0.64 | 49.13±0.15 |
| FedALA | 82.36±0.21 | 82.29±0.93 | 85.16±0.35 | 51.58±1.14 | 52.42±0.77 | 72.69±0.30 |
| GPFL | 74.42±1.03 | 73.03±1.56 | 76.49±0.48 | 45.02±0.40 | 50.14±0.26 | 72.75±0.84 |
| FedDBE | 73.31±0.84 | 75.95±0.04 | 77.74±0.92 | 46.59±0.52 | 49.58±0.75 | 73.12±0.34 |
| FedAS | 78.39±1.53 | 80.16±1.17 | 84.26±0.46 | 49.08±0.57 | 52.45±0.49 | 70.80±0.70 |
| FABLE | 83.83±0.05 | 84.92±0.14 | 85.90±0.13 | 51.56±1.03 | 54.24±0.41 | 75.24±0.09 |
| FABLE w/ linear | **84.27±0.02** | **85.31±0.12** | **85.93±0.15** | **52.99±0.32** | **54.59±0.50** | **77.40±0.14** |

## B.2 ADDITIONAL RESULT

To further validate the generalization of our method, we increase the client number to 50 and tested multiple Dirichlet distributions: Dir(0.01) and Dir(0.5). Additionally, we simulated a real-world scenario where only 10% of clients participate in each communication round. The result is shown in Table 8. As the experimental results show, our method remains effective under those broader settings, further demonstrating the general applicability of our approach.

## B.3 ADAPTABILITY ANALYSIS

Table 9: Performance comparison of various aggregation methods combining with FABLE. The best results are highlighted in **bold**, and the second-best results are underlined.

| | CIFAR 10 | | CIFAR 100 | | Sogou News | | AG News | |
|---|---|---|---|---|---|---|---|---|
| | IID | Non-IID | IID | Non-IID | IID | Non-IID | IID | Non-IID |
| FABLE | 85.45±0.26 | 93.78±0.11 | 57.15±0.04 | 70.73±0.40 | 94.99±0.04 | 98.39±0.02 | 91.42±0.16 | 97.43±0.08 |
| FABLE w/ linear | 85.90±0.10 | **94.34±0.35** | **59.91±0.09** | **73.25±0.06** | **95.26±0.03** | 98.46±0.03 | 91.95±0.19 | **97.95±0.03** |
| SCAFFOLD | 81.91±0.23 | 87.73±0.36 | 56.28±0.36 | 57.41±0.34 | 94.29±0.01 | 93.12±0.15 | 88.75±0.11 | 86.93±0.79 |
| SCAFFOLD w/ FABLE | 85.39±0.19 | 93.80±0.01 | 57.19±0.14 | 70.57±0.49 | 94.95±0.14 | 98.73±0.02 | 91.62±0.09 | 97.31±0.27 |
| SCAFFOLD w/ FABLE & linear | **85.93±0.10** | 94.21±0.11 | 59.80±0.23 | 73.09±0.14 | 94.97±0.03 | **98.76±0.02** | **92.20±0.12** | 97.38±0.04 |
| FedALA | 83.62±0.31 | 92.94±1.30 | 58.69±0.23 | 67.58±0.29 | 94.91±0.09 | 98.43±0.02 | 91.36±0.09 | 97.64±0.05 |
| FedALA w/ FABLE | 85.36±0.02 | 93.99±0.14 | 58.28±0.22 | 70.73±0.51 | 94.95±0.06 | 98.58±0.04 | 90.87±0.10 | 97.71±0.13 |
| FedALA w/ FABLE & linear | 85.75±0.04 | 94.11±0.08 | 59.71±0.38 | 73.13±0.08 | 95.05±0.03 | 98.60±0.03 | 91.90±0.03 | 97.73±0.07 |

FABLE focus on local training and remains orthogonal to server-side optimizations, enabling integration with classical federated aggregation strategies. In the main text, we integrate FABLE with the conventional FedAvg (McMahan et al., 2017). To further explore the adaptability, we integrate the FABLE anchor mechanism with SCAFFOLD (Karimireddy et al., 2020) and FedALA (Zhang et al., 2023c). The result is shown in Table 9. According to the results, we can see that aggregation strategies combing with FABLE shows improvements over origin origin performance across most data distributions, with notable gains in non-IID settings. It demonstrates the generalizability and flexibility of FABLE when combined with other aggregate methods. In future work, we hope to integrate FABLE with advanced server-side optimization techniques to further explore the balance between privacy protection and model performance.

## B.4 ANALYSIS ON PRIVACY PROTECTION

**Data Privacy.** We follow the experimental setup in Deep Leakage from Gradients (DLG) to evaluate the privacy risks under FABLE and other baselines. A modified version of ResNet-18 is em-

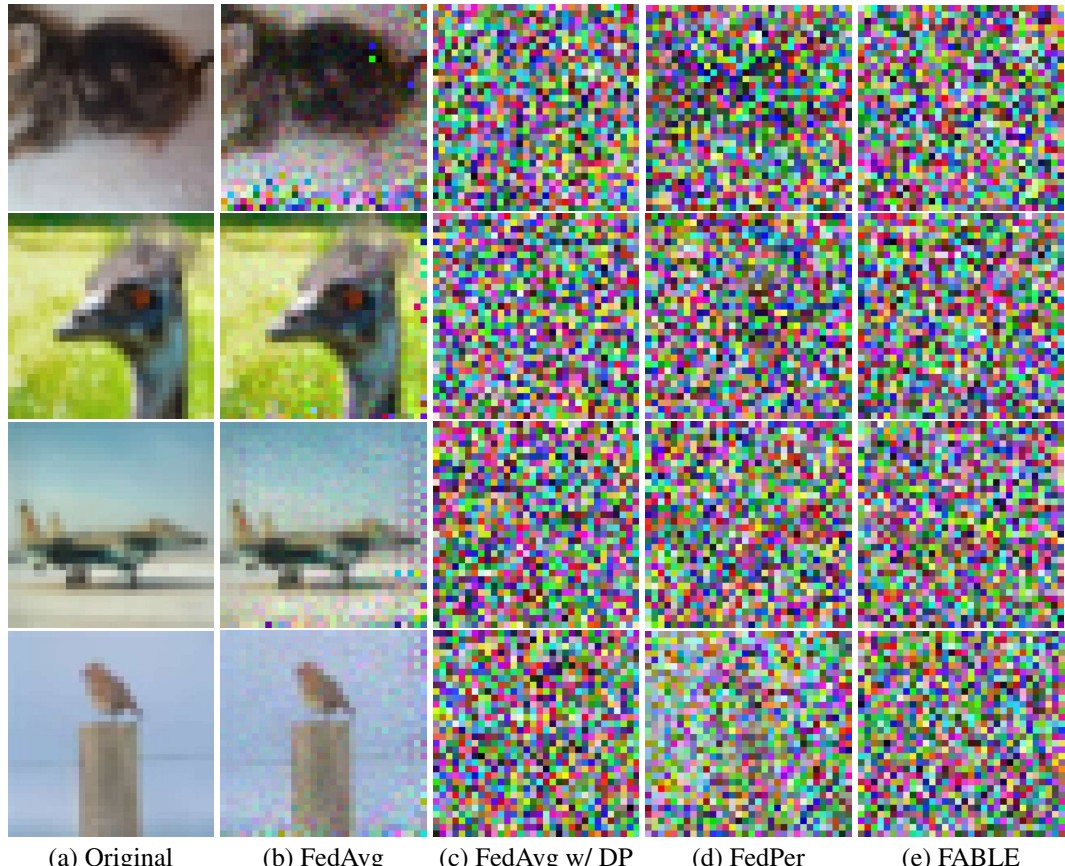

| (a) Original | (b) FedAvg | (c) FedAvg w/ DP | (d) FedPer | (e) FABLE |

Figure 5: Visualization of reconstructed images using DLG attack in different federated learning methods on CIFAR-10.

ployed for gradient leakage analysis. In this configuration, standard ReLU activations are replaced with Sigmoid functions to ensure second-order differentiability. Furthermore, all stride operations are removed to allow for finer-grained gradient updates. The optimization procedure utilizes the L-BFGS algorithm, with a total of 300 iterations conducted on the CIFAR-10 dataset. Additional visualization results are presented in Figure 5.

Table 10: Impact of anchor availability on model inference accuracy evaluated with different anchor numbers. The best results are highlighted in **bold**, and the second-best results are underlined.

| Anchors Numbers | 128 | | 256 | | 512 | | 1024 | |
| --- | --- | --- | --- | --- | --- | --- | --- | --- |
| | IID | Non-IID | IID | Non-IID | IID | Non-IID | IID | Non-IID |
| CIFAR-10 | **75.70±1.57** | **81.87±0.61** | 75.55±1.02 | 81.61±0.19 | 74.94±0.78 | 80.78±0.52 | 74.46±0.93 | 78.90±0.28 |
| CIFAR-100 | **56.76±0.17** | **67.13±0.93** | 56.20±0.31 | 66.42±1.21 | 55.96±0.38 | 66.10±0.43 | 54.99±0.43 | 65.76±0.49 |

**Model Privacy.** Our evaluation of model-level privacy follows the protocol outlined in MIA (Shokri et al., 2017). Specifically, we adopt the original MIA formulation, where an attack model is trained on the prediction outputs of shadow models, with each sample labeled as member or non-member based on its inclusion in the client training dataset. The attack model is implemented as a fully connected neural network and trained using stochastic gradient descent with cross-entropy loss. A key distinction in our setup lies in the construction of the shadow models: rather than training them from scratch, we exploit client-side model information exchanged during communication rounds. This approach provides a more realistic and practical threat model compared to the conventional shadow model paradigm in MIA.

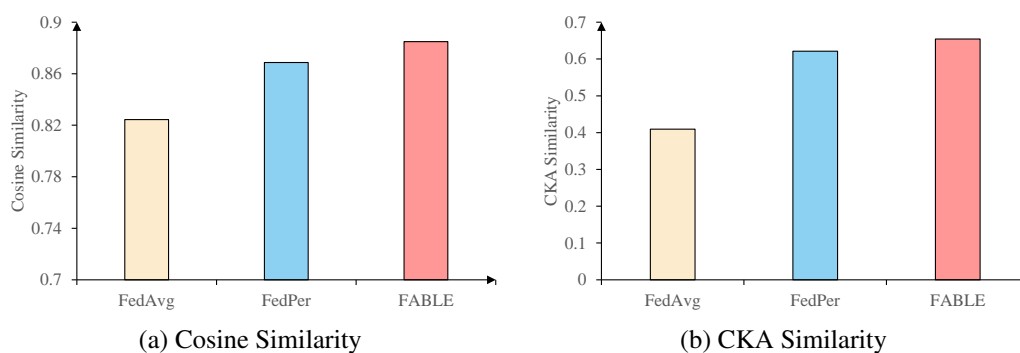

Figure 6: Comparison of representation similarity between the global encoder and client-specific local encoders across different FL methods on CIFAR-10 under the Non-IID setting. We measure the alignment of learned representations using (a) Cosine similarity and (b) Centered Kernel Alignment (CKA) similarity.

Table 11: Comparison of training overhead across different methods on CIFAR-10 and CIFAR-100.

|  | CIFAR-10 | | CIFAR-100 | |
|---|---|---|---|---|
|  | Memory(MB) | Time(s) | Memory(MB) | Time(s) |
| FedAvg | 4508 | 34.20±1.40 | 7190 | 42.67±0.49 |
| FedPer | 4508 | 33.92±0.40 | 7190 | 42.22±0.72 |
| FABLE w/ 128 Anchors | 4494 | 34.78±0.50 | 7188 | 43.65±1.59 |
| FABLE w/ 256 Anchors | 4576 | 35.11±1.01 | 7252 | 44.13±2.15 |
| FABLE w/ 512 Anchors | 4620 | 35.21±1.92 | 7396 | 44.33±0.88 |

Besides,we conduct additional experiments to explore the impact of the number of anchors on model privacy protection. We compare the impact of different numbers of anchors on the inference accuracy under the setting of model privacy in the original text, as shown in the Table 10. As observed from the results, with the increase in the number of anchors, the performance of the model declines slightly in terms of privacy. By controlling the number of anchors, we can find an trade-off between privacy protection and model performance.

## B.5 ANALYSIS ON REPRESENTATION ALIGNMENT

To evaluate the consistency of representations between the global and local encoders under heterogeneous data distributions, we conduct a representation alignment analysis. Specifically, we compute the representations of CIFAR-10 using the global encoder and each client's locally trained encoder under the Non-IID setting, then measure their pairwise similarity. We employ two widely used metrics including Cosine Similarity and Centered Kernel Alignment (CKA).

As shown in Figure 6, FABLE achieves the highest alignment scores across both similarity measures, outperforming FedAvg (McMahan et al., 2017) and FedPer (Arivazhagan et al., 2019). These results indicate that FABLE effectively mitigates the representation drift commonly observed in FL under non-IID conditions. By improving alignment between local and global encoders, FABLE reduces the risk of representation incompatibility during client local initialization phases. This leads to more stable local training and alleviates performance degradation caused by heterogeneous client updates.

## B.6 COMPLEXITY ANALYSIS

FABLE enhances data and model privacy by introducing anchor-aware representation transformations within a model-splitting-based PFL framework, while it incurs additional computational and memory overhead compared to classical FL methods. Specifically, unlike FedAvg (McMahan et al., 2017) and FedPer (Arivazhagan et al., 2019) which directly optimize the model via SGD on local

data, FABLE requires an intermediate representation transformation step to align local representations with fixed anchors. This efficient transformation inevitably introduces a marginal increase in training time.

Furthermore, to ensure anchor consistency across communication rounds, each client in FABLE must retain the full anchor set locally throughout the entire training process. To avoid redundant memory-to-GPU transfers at each round which would further increase the training cost, we choose to store anchors directly, rather than merely caching their indices. This design choice introduces a modest memory overhead, particularly as the number of anchors increases.

We quantitatively evaluate the global training cost under varying anchor number settings and compare it with other baselines, as shown in Table 11. Although FABLE exhibits slightly increased training time due to representation transformation, it achieves lower memory consumption with 128 anchors compared to FedAvg and FedPer. This reduction is attributed to the adaptive resizing of the decoder's input dimension, which decreases the number of model parameters when fewer anchors are used. When scaling up to 256 or 512 anchors, the additional overhead remains marginal and within acceptable practical limits. In real-world applications, the number of anchors can be flexibly adjusted to strike a balance between privacy protection, model generalization, and system efficiency.

## C    THE USE OF LARGE LANGUAGE MODELS (LLMS)

In the preparation of this manuscript, we utilized Large Language Models (LLMs) as a writing assistance tool. The role of the LLM was strictly limited to improving the language and readability of the text. Specific applications included correcting grammatical errors, refining sentence structure for clarity and flow, and polishing the overall prose.

