# OpenReview forum: "FABLE: Federated Anchor-Based Learning with Privacy Protection"
_ICLR.cc/2026/Conference — Submitted to ICLR 2026_

### Official Review · Reviewer_JkcM · 2025-10-27

**Soundness:** 2
**Presentation:** 3
**Contribution:** 2
**Rating:** 4
**Confidence:** 4

**Summary:**

This paper proposes an algorithm to protect privacy and handle heterogeneous data in federated learning. It proposes a globally shared encoder (trained using FedAvg) consisting of the earlier layers of the encoder. Each client maintains a private decoder with a classifier head. The novelty in the paper is that each client transforms the embedding vectors from the global encoder using a random projection and uses them as input to the decoder. As the projection matrix is private and random, and the decoder is private, this preserves privacy.

**Strengths:**

1. The problems of privacy and data heterogeneity are important in Federated Learning.
2. The paper is overall well-written and easy to read.

**Weaknesses:**

Weakness:
1. Why the anchor-based strategy would work is unclear. While projecting to a random coordinate system preserves pairwise distances approximately (JL lemma), it only happens when multiple projections are considered using multiple random coordinate systems. However, here only one system is used for each client. Can the authors show the insight using a simple setting so that the readers can have confidence in the method?
2. As the public encoder layers are situated near the input, their parameters contain information about the input data. So, they will expose some sort of private data.
3. How do you choose the anchor points? The results in Table 7 are confusing. The anchors seem to be fixed for the whole duration of the training; otherwise, the decoder training won't converge. So, in that case, what is the meaning of k-means clustering and fps, on which point cloud are these done?
4. It is unclear what the meaning of the linear transformation (Eq 4). I'm unclear why it is needed? One can use the dot product without normalization as the transformation. How is W_k initialized in practice? Does the finally learned W_k actually recover the dot products?
5. What the non-IID setting is unclear.
6. The use of anchors as secret keys is an interesting aspect, which, unfortunately, is too brief. The authors may want to build on this with formal guarantees to make the privacy argument solid.
7. Which component of the algorithm addresses non-iid data is unclear. If the data is too skewed, layers near the input will have different parameter values across clients (think clients have disjoint classes). Then the simple average of the parameters is known not to be a good proxy, so why does the simple averaging of the earlier layers work in non-iid setting is unclear.
8. Couldn't quite understand the difference between the data and model privacy in this context. Though there is a description in the paper, it'll be good to clarify it further.

**Questions:**

As above.

---

> ### Author Response · Authors · 2025-11-26
>
> We sincerely thank you for your thoughtful and detailed review. We hope to address all your concerns below.
>
> ---
> Weakness 1
>
> Why the anchor-based strategy would work is unclear. While projecting to a random coordinate system preserves pairwise distances approximately (JL lemma), it only happens when multiple projections are considered using multiple random coordinate systems. However, here only one system is used for each client. Can the authors show the insight using a simple setting so that the readers can have confidence in the method?
>
> ---
> **Answer:**
>
> **Our method's intuition is based on learning in a personalized relationship space, not on the JL lemma.** We appreciate you seeking a deeper understanding for our work. You are correct that our method is not directly based on the JL lemma. Instead, it is motivated by a different geometric intuition that similar samples have similar relative positions to a given sample in the representation space. We transform the learning problem by representing each data point not by its absolute coordinates, but by its representation of similarities to a client's private set of anchors. This preserves local geometric structure while allowing each client to learn a coordinate system optimally suited for its local task.
>
> ---
> Weakness 2
>
> As the public encoder layers are situated near the input, their parameters contain information about the input data. So, they will expose some sort of private data.
>
> ---
> **Answer:**
>
> **We agree that model-splitting alone provides limited privacy, which is why FABLE's anchor transformation is a necessary additional defense.** As our own results in Figure 2 demonstrate, a simple model-splitting approach still allows an attacker to partially reconstruct private data. The core of our privacy protection is not just the model split, but the gradient obfuscation mechanism. The mechanism entangles the gradients with the client's secret anchors, effectively "encrypting" them and rendering standard gradient inversion attacks ineffective  even if the encoder is known. FABLE's protection is thus a much stronger defense built on the basic model-splitting framework.
>
> ---
> Weakness 3
>
> How do you choose the anchor points? The results in Table 7 are confusing. The anchors seem to be fixed for the whole duration of the training; otherwise, the decoder training won't converge. So, in that case, what is the meaning of k-means clustering and fps, on which point cloud are these done?
>
> ---
> **Answer:**
>  * **The anchor selection is a one-time operation at initialization to ensure training stability.** We clarify that for all strategies, the anchor set is chosen only once at the beginning of the FL process and remains fixed. For k-means and fps, we first generate a representation set from the client's entire local dataset using a encoder, and then apply the selection algorithm. This process does not affect decoder convergence.
> *   **FABLE is robust to the anchor selection strategy, so we default to the most efficient one.** As shown in Table 7, the performance is very similar across random, k-means, and fps. This robustness allows us to use the computationally cheapest "random" strategy in our main experiments without sacrificing performance.
>
> ---
> Weakness 4
>
> It is unclear what the meaning of the linear transformation (Eq 4). I'm unclear why it is needed? One can use the dot product without normalization as the transformation. How is W_k initialized in practice? Does the finally learned W_k actually recover the dot products?
>
> ---
> **Answer:**
> * **The linear transformation $W_k$ is a adapter on the angle-based representation from $T_k$.** Our transformation based on cosine similarity preserves robust geometric patterns but discards scale information, which can make the downstream classification task more challenging. $W_k$ is a learnable and personalized layer that compensates for this by projecting the representation into a new space that is optimal for the downstream decoder, compensating for the lost scale information.
> * **Our disentangled design of using cosine similarity plus a linear layer is more robust than a simple dot product.** A dot product couples angle and magnitude, our approach separates them. This prevents large variations in representation norms across heterogeneous clients from destabilizing the training of the shared encoder, leading to a more robust design. $W_k$ is randomly initialized and learns to minimize the classification loss, not necessarily to recover the original dot products.

---

> ### Author Response · Authors · 2025-11-26
>
> ---
> Weakness 5
>
> What the non-IID setting is unclear.
>
> ---
> **Answer:**
>
> **We use a standard Dirichlet distribution to create non-IID data partitions, and FABLE's advantage grows with the degree of data skew.** We followed a standard protocol in common FL, using a Dirichlet distribution to partition data by label distribution across the 20 or 50 clients. We provide a detailed description in the appendix and also experiment with various skewness. Our results show that FABLE's relative performance advantage over baselines becomes even more pronounced under more extreme data skew.
>
> ---
> Weakness 6
>
> The use of anchors as secret keys is an interesting aspect, which, unfortunately, is too brief. The authors may want to build on this with formal guarantees to make the privacy argument solid.
>
> ---
> **Answer:**
> * **We mathematically formalize the formal guarantee via the Rank-Nullity Theorem.** We model the gradient generation process as a Markov Chain and analyze the Mutual Information $I(X; G)$. We prove that when the number of anchors $N$ is smaller than the feature dimension $D$, the anchor-aware transformation matrix becomes rank-deficient. According to the Rank-Nullity Theorem, this induces a non-trivial Null Space of dimension $D-N$. Any gradient information aligning with this Null Space is physically annihilated during backpropagation. This creates an irreversible information-theoretic bottleneck, ensuring strictly lower leakage compared to baselines $I(X; G_{\text{FABLE}}) < I(X; G_{\text{base}})$.
> * **We further formalize the role of anchors as keys by analyzing the scenario with more anchors.** Even in cases where the anchor set is large $N \ge D$, we utilize the Data Processing Inequality (DPI) to prove that FABLE introduces no additional leakage compared to the baseline $I(X; G_{\text{FABLE}}) \le I(X; G_{\text{base}})$. Without access to $\mathcal{A}$, the adversary faces a high conditional entropy barrier, making precise gradient inversion an ill-posed problem.
>
> ---
> Weakness 7
>
> Which component of the algorithm addresses non-iid data is unclear. If the data is too skewed, layers near the input will have different parameter values across clients (think clients have disjoint classes). Then the simple average of the parameters is known not to be a good proxy, so why does the simple averaging of the earlier layers work in non-iid setting is unclear.
>
> ---
> **Answer:**
>
> **FABLE addresses non-IID data by decomposing the learning task: the shared encoder learns general features while private components handle personalization.** The public encoder learns to extract general and transferable features, which remain meaningful to average even across skewed distributions. The personalization is handled by the private components including decoder, $W_k$, and anchors. Our anchor transformation encourages the shared encoder to learn a representation space where clients are more aligned in terms of geometric structure, as supported by our representation alignment analysis in the appendix.
>
> ---
> Weakness 8
>
> Couldn't quite understand the difference between the data and model privacy in this context. Though there is a description in the paper, it'll be good to clarify it further.
>
> ---
> **Answer:**
>
> **We clarify the distinction between data privacy and model privacy.** Data privacy aims to protect the raw training data from being reconstructed from model updates, which we evaluate via DLG attacks. Model privacy aims to protect the trained model itself from unauthorized use or intellectual property theft. We evaluate this by showing that even if an attacker steals the full model parameters, the model is useless without the private anchor as demonstrated in our MIA and inference-without-anchors experiments
>
> ---
> We hope these clarifications address the reviewer’s concerns. If you have any follow-up question, we are more than happy to clarify.

---

### Official Review · Reviewer_xL6J · 2025-10-29

**Soundness:** 2
**Presentation:** 2
**Contribution:** 2
**Rating:** 2
**Confidence:** 3

**Summary:**

This paper proposes FABLE (Federated Anchor-Based Learning), a personalized federated learning (PFL) method that aims to jointly address privacy leakage and data heterogeneity in federated learning (FL). Each client selects a small set of private anchors from its local dataset and uses them to perform a client-specific representation transformation before model training. These anchors are never shared, which is claimed to provide dual privacy protection—preventing data reconstruction attacks (e.g., DLG) and protecting model intellectual property (via “anchor-dependent” inference). FABLE is evaluated on CIFAR-10, CIFAR-100, AG News, and Sogou News under IID and non-IID settings, showing performance competitive with or slightly better than several FL and PFL baselines. Experiments also include analyses of privacy (PSNR, MIA accuracy), ablations on anchor numbers, selection strategies, and linear transformation variants.

**Strengths:**

- Privacy evaluation includes both data-level (DLG) and model-level (MIA) metrics, which is appreciated.
- The additional linear transformation is computationally lightweight and improves model stability.

**Weaknesses:**

- The “private anchor” idea is positioned as novel, but the technical distinction (anchors not being shared) is incremental rather than conceptually transformative.
- The privacy claims (especially model privacy via anchors as “secret keys”) are largely qualitative. The analysis relies on intuition (e.g., Equations (6)-(10)) without rigorous quantification or formal privacy guarantees (e.g., DP bounds, mutual information reduction).
- No convergence or generalization analysis is provided for FABLE, despite this being standard in FL papers. It remains unclear how the transformation affects optimization stability or gradient variance in theory.
- The privacy–utility tradeoff is not clearly quantified. The performance improvements over strong PFL baselines (e.g., FedALA, GPFL, FedDBE) are modest.

**Questions:**

1. Beyond not sharing anchors, does FABLE introduce any new theoretical insight or architectural mechanism that could be considered conceptually distinct?
2. The privacy analysis (Eqs. 6-10) provides an intuitive description of how anchors obscure gradients. Can the authors provide a formal privacy guarantee (e.g., differential privacy bounds, mutual-information analysis, or formal proof of obfuscation strength)? How sensitive is the proposed privacy protection to the number or distribution of anchors? For example, does a small anchor set materially weaken protection?
3. The privacy–utility trade-off is qualitatively described but not quantitatively analyzed. Could the authors provide a more explicit evaluation, e.g., performance versus privacy level (PSNR or MIA accuracy) curves?
4. The improvements over strong PFL baselines (FedALA, GPFL, FedDBE) appear modest. Could the authors clarify whether these gains are statistically significant? How would FABLE perform on more complex models or real-world FL datasets to demonstrate scalability?

---

> ### Author Response · Authors · 2025-11-26
>
> We sincerely thank you for your critical and detailed review of our manuscript. We hope to address your concerns and clarify the novelty and significance of FABLE.
>
> ---
> Weakness 1
>
> The “private anchor” idea is positioned as novel, but the technical distinction (anchors not being shared) is incremental rather than conceptually transformative.
>
> Question 1
>
> Beyond not sharing anchors, does FABLE introduce any new theoretical insight or architectural mechanism that could be considered conceptually distinct?
>
> ---
> **Answer:**
> * **Our key conceptual innovation is redefining the role of anchors in FL from a public alignment tool to a private key.** Unlike prior work where public anchors serve to align features across clients, FABLE's private anchors create a unique, personalized coordinate system for each client. The privacy of the anchors is not just a feature but the foundational principle enabling both dual privacy protection and deep personalization.
> * **FABLE introduces a novel architectural mechanism where a single component provides both personalization and privacy obfuscation.**  We explicitly construct a private coordinate system via the anchor-based transformation. To our knowledge, this mechanism serves a dual purpose that is novel. It personalizes the model by creating a client-specific representation, and obfuscates the gradients by entangling them with the secret anchors. This dual-role mechanism is a conceptually distinct contribution, and we believe this represents a conceptually novel direction for federated learning.
>
> ---
> Weakness 2
>
> The privacy claims (especially model privacy via anchors as “secret keys”) are largely qualitative. The analysis relies on intuition (e.g., Equations (6)-(10)) without rigorous quantification or formal privacy guarantees (e.g., DP bounds, mutual information reduction).
>
> Question 2
>
> The privacy analysis (Eqs. 6-10) provides an intuitive description of how anchors obscure gradients. Can the authors provide a formal privacy guarantee (e.g., differential privacy bounds, mutual-information analysis, or formal proof of obfuscation strength)? How sensitive is the proposed privacy protection to the number or distribution of anchors? For example, does a small anchor set materially weaken protection?
>
> ---
> **Answer:**
> - **We have derived a formal privacy guarantee based on Information Theory, proving that FABLE reduces information leakage compared to standard FedAvg.** We formally define the gradient leakage as the Mutual Information $I(X; G)$ between private data $X$ and exposed gradients $G$. We prove that when the number of anchors $N$ is smaller than the original feature dimension $D$, the transformation matrix becomes rank-deficient. According to the Rank-Nullity Theorem, this induces a non-trivial Null Space of dimension $D-N$. Any gradient information aligning with this Null Space is physically annihilated. This creates an irreversible information loss, guaranteeing that $I(X; G_{\text{FABLE}}) < I(X; G_{\text{base}})$.
> - **We prove non-degradation and obfuscation strength for larger anchor sets.** Even in cases where the anchor set is large $N \ge D$, we utilize the Data Processing Inequality (DPI) to prove that FABLE introduces no additional leakage compared to the baseline $I(X; G_{\text{FABLE}}) \le I(X; G_{\text{base}})$. Without access to $\mathcal{A}$, the adversary faces a high conditional entropy barrier, making precise gradient inversion an ill-posed problem.
> - **Sensitivity analysis confirms that fewer anchors enhance privacy protection.** Our theoretical analysis suggests a smaller $N$ imposes a bottleneck. We evaluated model privacy where an attacker possesses model parameters but lacks anchors, as shown in Table 10. The results show that as $N$ increases, the strength of model privacy protection slightly decrease. This empirical trend aligns with our derived theoretical bound, where the dimension of the information loss $D-N$ is maximized at lower $N$. This confirms that smaller anchor sets provide stronger protection by strictly limiting the rank of the exposed information.

---

> ### Author Response · Authors · 2025-11-26
>
> ---
> Weakness 3
>
> No convergence or generalization analysis is provided for FABLE, despite this being standard in FL papers. It remains unclear how the transformation affects optimization stability or gradient variance in theory.
>
> ---
> **Answer:**
> * **The anchor-based transformation does not disrupt stability because it acts as a fixed, differentiable, and smooth layer.** We prove that under the standard assumption of bounded feature norms, the cosine-similarity-based transformation $\mathcal T_k(\cdot; \mathcal{A}_k)$ satisfies the Lipschitz smoothness condition. This ensures that the local objective functions remain $L$-smooth, providing the necessary mathematical guarantee.
> * **FABLE achieves the standard convergence rate of $\mathcal{O}(1/\sqrt{T})$ comparable to FedAvg.** Our derivation confirms that  the algorithm converges to a first-order stationary point with the same efficiency order as standard non-convex FedAvg. This theoretically validates that FABLE’s privacy and personalization benefits do not come at the cost of a slower convergence order. We will include these details in the revised manuscript to strengthen the theoretical part.
>
> ---
> Question 3
>
> The privacy–utility trade-off is qualitatively described but not quantitatively analyzed. Could the authors provide a more explicit evaluation, e.g., performance versus privacy level (PSNR or MIA accuracy) curves?
>
> ---
> **Answer:**
>
> **We will add a new privacy-utility trade-off plot to quantitatively visualize FABLE's superior balance.** We already have several privacy proxies and performance metrics and they are currently presented in tabular. We promise that these will be combined into curves.
>
> ---
> Weakness 4
>
> The privacy–utility tradeoff is not clearly quantified. The performance improvements over strong PFL baselines (e.g., FedALA, GPFL, FedDBE) are modest.
>
> Question 4
>
> The improvements over strong PFL baselines (FedALA, GPFL, FedDBE) appear modest. Could the authors clarify whether these gains are statistically significant? How would FABLE perform on more complex models or real-world FL datasets to demonstrate scalability?
>
> ---
> **Answer:**
> * **FABLE demonstrates more substantial gains in the more challenging and realistic cross-device setting.** While gains are moderate in the cross-silo experiments in the main text, our results in the appendix Table 8 show a more significant performance advantage over strong PFL baselines in a cross-device simulation, demonstrating its scalability and effectiveness.
> * **FABLE's main contribution is its superior position on the privacy-utility trade-off, which existing PFL baselines cannot achieve without catastrophic performance loss.** Strong PFL methods have no additional privacy while adding it via DP would decimate their performance. FABLE provides high utility and strong privacy simultaneously.
> * **Our approach is orthogonal to and can be combined with advanced server-side optimizations.** As shown in our adaptability analysis Table 9, FABLE's client-side mechanism can be integrated with methods like SCAFFOLD to achieve even better performance, highlighting its flexibility.
>
> ---
> We hope these clarifications fully address the reviewer’s concerns and contribute to the overall strength of the paper. If you have any follow-up question, we are more than happy to clarify.

---

### Official Review · Reviewer_nruk · 2025-10-30

**Soundness:** 2
**Presentation:** 2
**Contribution:** 2
**Rating:** 4
**Confidence:** 3

**Summary:**

The paper proposes FABLE (Federated Anchor-Based Learning), a personalized FL method that introduces \emph{private, client-local anchors} to address both privacy leakage from shared encoder gradients and performance degradation under non-IID data. Each client $k$ keeps an anchor set $A_k \subset D_k$, embeds both its data and anchors with the shared/global encoder $g_k(\cdot)$, and transforms an input $x$ into an \emph{anchor-aware} representation
$$
r_k(x) = T_k(g_k(x); A_k) =
\big( \cos\big(g_k(x), g_k(a_1^k)\big), \dots, \cos\big(g_k(x), g_k(a_{|A_k|}^k)\big) \big),
$$
optionally followed by a client-specific linear layer
$\hat r_k(x) = W_k \, r_k(x)$.
The transformed feature is then fed to a private decoder. Since the server only sees gradients that are \emph{already} passed through the unknown, private anchor transform, the paper argues that DLG-style gradient inversion becomes ill-posed. Experiments on CIFAR-10/100 and text datasets under IID and Dirichlet($0.1$) non-IID splits show that FABLE (especially with the linear layer) matches or outperforms strong PFL baselines while offering better resistance to reconstruction attacks.

**Strengths:**

Originality: The core idea is to make the client work in its own, anchor-conditioned coordinate system that is never shared. Unlike approaches that distribute public/synthetic anchors, FABLE keeps $A_k$ private and lets $A_k$ directly modulate what the server ever sees. This neatly ties together personalization and privacy.

Quality: The method is concretely specified: split model into shared encoder and private decoder; form anchor similarities via cosine, freeze anchor embeddings for stability, and add a local linear map $W_k \in \mathbb{R}^{d \times d}$ to restore expressiveness. The privacy discussion explicitly rewrites gradient leakage as inverting a composite mapping $F_k(\cdot; A_k)$ rather than the plain encoder.

Clarity: The dataflow in Fig.1 corresponds directly to Eqs.(2)-(4): global encoder $\to$ anchor similarity $\to$ (optional) linear transform $\to$ private head. Ablations on the number of anchors (256, 512, 1024), on the selection strategy (random, k-means, FPS), and on the linear layer make the contribution easy to verify.

Significance: The result that removing anchors at inference time makes the model basically unusable (accuracy drops to near-random) is practically meaningful: the model is effectively bound to the client’s private anchors, providing a concrete form of model-level privacy in FL deployments.

**Weaknesses:**

- Privacy claim is informal. The paper argues that the server now observes a gradient of the form $
  \nabla_{\theta_g} \mathcal{L}\big( F_k(x; A_k), y \big)$, where $F_k(\cdot; A_k)$ is the anchor-induced transform, and that an attacker cannot run DLG because $A_k$ is unknown. However, there is no quantitative bound, and no experiment where the attacker jointly optimizes over a guessed anchor set. This makes the privacy result more suggestive than proved.

- Anchor secrecy is a single point of failure. The “dual privacy” claim assumes the attacker can get model parameters but not the anchors. In realistic client compromise, $A_k$ is just local data and can be exfiltrated, in which case the protection collapses. The paper should state this assumption clearly.

- High-dimensional anchor spaces. With $|A_k| = 512$, the representation $r_k(x) \in \mathbb{R}^{512}$ and the client learns a $512 \times 512$ matrix $W_k$. This is fine for CIFAR and 20 clients, but the paper does not profile the compute/comm overhead for larger models, more clients, or bigger anchor sets, despite claiming “no significant overhead.”

- Comparison to other anchor-style FL is thin. The main difference from existing anchor/synthetic-prototype FL is that FABLE does not share its anchors. A head-to-head with a baseline that uses the same transform but with public anchors would isolate how much of the gain is from secrecy vs.\ just better alignment.

- Gains over strong PFL baselines are modest. On the hardest non-IID vision setting, FABLE w/ linear is best but only by a couple of percentage points over FedALA / FedPer. This slightly weakens the “one method fixes personalization and privacy” message.

**Questions:**

1- Joint inversion attack. Your privacy discussion effectively assumes an attacker solves $\min_{x', y'} \left\| \nabla_{\theta_g} \mathcal{L}\big( F_k(x'; A_k), y' \big) - G_{\text{obs}} \right\|^2$ without knowing $A_k$. Did you try the more realistic attack $\min_{x', y', A_k'} \left\| \nabla_{\theta_g} \mathcal{L}\big( F_k(x'; A_k'), y' \big) - G_{\text{obs}} \right\|^2$, i.e. treating $A_k$ as latent variables? This would directly test how much privacy comes from anchor secrecy.

2- Anchor representation. Are anchors stored and used as \emph{raw} samples $a_i^k \in D_k$, or as encoded features $g_k(a_i^k)$ cached once? If raw samples are used, do you rotate/refresh them to prevent slow leakage?

3- Dimensional mismatch. You set $|A_k| = d = 512$. What happens if the encoder output dimension $d_{\text{enc}} \neq |A_k|$? Do you use a rectangular $W_k \in \mathbb{R}^{d_{\text{out}} \times |A_k|}$, and does the privacy effect survive this mismatch?

4- Failure without anchors. In Fig. 3, accuracy collapses when anchors are removed. Is this because $r_k(x)$ becomes almost constant (low cosine similarities to non-existent anchors), or because $W_k$ was trained on a specific anchor distribution that is now missing? Showing the empirical distribution of $r_k(x)$ with/without anchors would clarify.

---

> ### Author Response · Authors · 2025-11-26
>
> We sincerely thank you for your thorough and insightful review.  We appreciate your praise for the originality and importance of our work and are grateful for the opportunity to address your questions.
>
> ---
> Weakness 1
>
> Privacy claim is informal. The paper argues that the server now observes a gradient of the form$\nabla_{\theta_g} \mathcal{L}(F_k(x; A_k), y),$ where $F_k(\cdot; A_k)$ is the anchor-induced transform, and that an attacker cannot run DLG because $A_k$ is unknown. However, there is no quantitative bound, and no experiment where the attacker jointly optimizes over a guessed anchor set. This makes the privacy result more suggestive than proved.
>
> Question 1
>
> Joint inversion attack. Your privacy discussion effectively assumes an attacker solves $\min_{x’, y’, A_k’} \left| \nabla_{\theta_g} \mathcal{L}(F_k(x’; A_k’), y’) - G_{\text{obs}} \right|^2$ without knowing $A_k$. Did you try the more realistic attack $\min_{x’, y’, A_k’} \left| \nabla_{\theta_g} \mathcal{L}(F_k(x’; A_k’), y’) - G_{\text{obs}} \right|^2$, i.e. treating $A_k$ as latent variables? This would directly test how much privacy comes from anchor secrecy.
>
> ---
> **Answer:**
> * **New experiments demonstrate that jointly optimizing anchors actually degrades attack performance, confirming FABLE’s robustness against sophisticated inversion.** Following your suggestion, we extended the DLG to treat anchors as learnable latent parameters. As shown in the table below, this joint optimization strategy resulted in a lower PSNR compared to the vanilla DLG baseline. While this may seem counter-intuitive, it highlights the computational security of FABLE. By treating anchors as variables, the attacker exponentially expands the search space. The optimizer exploits these additional degrees of freedom to minimize the gradient matching loss by generating noise anchors that compensate for gradient discrepancies, rather than converging toward the true private data.
>
> |                                            | PSNR          |
> | ------------------------------------------ | ---------------- |
> | FABLE (DLG + Anchor Recon.)                      | 13.39 $\pm$ 1.12 |
> | FABLE (Vanilla DLG)                        | 14.81 $\pm$ 0.90 |
> * **We have formalized the privacy guarantee using Information Theory to prove a reduction in accessible information.** We prove that the information an attacker gains from the observed gradient $G'$ is less than that from the original gradient $G$: **$I(X; G') < I(X; G)$**. This inequality holds because the residual information term $I(X; G | G')$ is positive. This residual information stems from two sources: (1) Information Loss, where gradient components in the null space of the anchor-induced transformation are irreversibly discarded  and (2) Key Ambiguity, where a vast ambiguity set of potential original gradients maps to the same observed gradient due to the secrecy of the anchor set $\mathcal{A}$.
>
> ---
> Weakness 2
>
> Anchor secrecy is a single point of failure. The “dual privacy” claim assumes the attacker can get model parameters but not the anchors. In realistic client compromise, $A_k$ is just local data and can be exfiltrated, in which case the protection collapses. The paper should state this assumption clearly.
>
> ---
> **Answer:**
> * **FABLE's privacy claims are conditioned on the standard FL threat model of an honest-but-curious server and passive adversaries, with trusted and uncompromised clients.** Regarding the threat model and the role of anchor secrecy as a single point of failure, we agree with your assessment. Our primary contribution is to defend against the prevalent threat of privacy leakage from communicated model updates. Under this assumption, FABLE protects data privacy by resisting gradient inversion attacks and protects model privacy by preventing unauthorized use of the model. We will make this assumption and the scope of our threat model explicitly clear in the revised manuscript.
> * **We do not claim protection against a complete compromise of a client in which local data and anchors are stolen.** This represents a worst case scenario that is generally considered indefensible in most federated learning settings. If a client is fully compromised, the attacker will naturally obtain all of its local data, including the samples chosen as anchors. In such a situation, FABLE becomes ineffective.

---

> ### Author Response · Authors · 2025-11-26
>
> ---
> Weakness 3
>
> High-dimensional anchor spaces. With $|A_k| = 512$, the representation $r_k(x) \in \mathbb{R}^{512}$ and the client learns a $512 \times 512$ matrix $W_k$. This is fine for CIFAR and 20 clients, but the paper does not profile the compute/comm overhead for larger models, more clients, or bigger anchor sets, despite claiming “no significant overhead.”
>
> ---
> **Answer:**
>
> **FABLE's communication overhead is identical to model-splitting baselines, and local compute costs are manageable and tunable.** Compared with approaches that transmit the full model in each round such as FedAvg, FABLE requires less communication because anchors and the private decoder stay on the client side. Our complexity analysis in the appendix shows a clear tradeoff in which using a smaller anchor set lowers local computation and memory overhead while causing only a slight decrease in accuracy. This design allows each client to adjust the overhead according to its own resource constraints and performance requirements.
>
> ---
> Weakness 4
>
> Comparison to other anchor-style FL is thin. The main difference from existing anchor/synthetic-prototype FL is that FABLE does not share its anchors. A head-to-head with a baseline that uses the same transform but with public anchors would isolate how much of the gain is from secrecy vs.\ just better alignment.
>
> ---
> **Answer:**
> * **We conducted the additional experiment you suggested, and the results show that using public anchors improves the performance of FABLE.** Following your insightful advice, we implemented "FABLE-Public", which relies on shared public anchors. The results are reported in the table below. We argue that the performance gains of public anchors over the privacy anchor based FABLE arise from better alignment of representations.
>
> |                        | CIFAR-10 |         | CIFAR-100 |         |
> | ---------------------- | -------- | ------- | --------- | ------- |
> |                        | IID      | Non-IID | IID       | Non-IID |
> | FABLE w/ public anchor | 85.55    | 93.82   | 57.30     | 70.98   |
> | FABLE                  | 85.45    | 93.78   | 57.15     | 70.73   |
> * **This experiment further demonstrates that FABLE adds strong privacy at virtually no cost to model utility.** The key finding is that our original FABLE with private anchors attains accuracy comparable to its public variant while also providing additional privacy protection relative to "FABLE-Public". This observation directly supports our central claim that FABLE offers an excellent privacy utility tradeoff by integrating strong privacy without paying the performance degradation price typical of other privacy techniques like Differential Privacy.
>
> ---
> Weakness 5
>
> Gains over strong PFL baselines are modest. On the hardest non-IID vision setting, FABLE w/ linear is best but only by a couple of percentage points over FedALA / FedPer. This slightly weakens the “one method fixes personalization and privacy” message.
>
> ---
> **Answer:**
>
> **FABLE demonstrates more substantial gains in the more realistic cross-device setting and offers an balance of privacy and utility.** While gains are moderate in the cross-silo setting, our results in Table 8 show a more significant advantage in a cross-device setting. More importantly, FABLE's primary contribution is not just to advance SOTA in accuracy, but to do so while providing strong privacy for free. Unlike baselines that would suffer catastrophic performance degradation if combined with DP, FABLE achieves a superior trade-off between privacy and utility. Furthermore, as shown in Table 9, FABLE is orthogonal to and can be combined with other server-side optimizations.
>
> ---
> Question 2
>
> Anchor representation. Are anchors stored and used as raw samples $a_i^k \in D_k$, or as encoded features $g_k(a_i^k)$ cached once? If raw samples are used, do you rotate/refresh them to prevent slow leakage?
>
> ---
> **Answer:**
>
> **Anchors are stored as raw samples and their features are computed dynamically using the latest global encoder from the server.** We use the latest encoder $g_k$ from the server to compute anchor features at the start of each local epoch. The anchor set itself remains fixed to ensure training stability and maintain secrecy. A stop-gradient is used on these features during backpropagation.

---

> ### Author Response · Authors · 2025-11-26
>
> ---
> Question 3
>
> Dimensional mismatch. You set $|A_k| = d = 512$. What happens if the encoder output dimension $d_{\text{enc}} \ne |A_k|$? Do you use a rectangular $W_k \in \mathbb{R}^{d_{\text{out}} \times |A_k|}$, and does the privacy effect survive this mismatch?
>
> ---
> **Answer:**
> * **Our framework is flexible to dimensional mismatches between the encoder output and the number of anchors.** The case of $|A_k| = d_\text{enc}$ is for simplicity. The core mechanism for computing the private representation $r_k$ only requires that the data samples and the anchors are projected by the encoder into the same feature space of dimension $d_\text{enc}$. The number of anchors $|A_k|$ solely determines the dimensionality of the subsequent private representation $r_k$, not the output dimension of the encoder. To handle this, the input layer of the private decoder is dynamically adapted to match the dimension $|A_k|$. As shown in our complexity analysis in the appendix, this design choice even allows for a reduction in memory overhead when a smaller anchor set is used, as the decoder becomes more lightweight.
> * **The privacy effect is robust and survives this mismatch.** The core privacy mechanism stems from the fact that the server observes a gradient that has been transformed by an unknown, client-specific mapping induced by the secret anchor set A_k. This obfuscation is a structural property of our method and does not depend on whether $|A_k|$ equals $d_\text{enc}$. Furthermore, the analysis of model privacy in Table 10 evaluates the impact of varying the number of anchors, was conducted with a fixed encoder dimension. These experiments operate in the $|A_k| ≠ d_\text{enc}$ setting you asked about and show that the privacy protection remains strong across all configurations.
>
> ---
> Question 4
>
> Failure without anchors. In Fig. 3, accuracy collapses when anchors are removed. Is this because $r_k(x)$ becomes almost constant (low cosine similarities to non-existent anchors), or because $W_k$ was trained on a specific anchor distribution that is now missing? Showing the empirical distribution of $r_k(x)$ with/without anchors would clarify.
>
> ---
> **Answer:**
>
> **The performance collapse without anchors is caused by a severe out-of-distribution shift, confirming the model's dependency on the private anchors.** The collapse is indeed a result of a severe distribution shift. When the anchor transformation $T_k$ is removed during inference, the input to the subsequent layers is no longer the angle-based representation it was trained on. This out-of-distribution input naturally leads to a catastrophic failure in prediction performance, confirming that the model's functionality is deeply entangled with the presence of the corresponding private anchors.
>
> ---
> Thank you once again for your meticulous and valuable feedback. If you have any follow-up question, we are more than happy to clarify.

---

> > ### Comment · Reviewer_nruk · 2025-11-26
> >
> > Thank you for the response and clarifications.
> >
> > You show PSNR, but is it possible to show SSIM/success-rate and results accross seeds?
> > Perhaps a proof sketch (assumptions and where the residual term comes from) could be added.
> > A comparison for Client cost vs. |A| (FLOPs/mem/time) and round time vs #clients. Currently, these are stated, not measured.
> >
> > I am maintaining my current recommendation.

---

> > > ### Author Response · Authors · 2025-12-04
> > >
> > > We sincerely thank you for your constructive feedback. We have conducted additional experiments across random seeds, formalized the privacy proof, and performed rigorous benchmarking of computational costs.
> > >
> > > ---
> > > You show PSNR, but is it possible to show SSIM/success-rate and results across seeds?
> > >
> > > ---
> > >
> > > * **New experiments across 3 random seeds confirm FABLE’s consistent robustness measured by SSIM and Success Rate.** We define success rate as the probability that a pre-trained ResNet-18 correctly identifies the label of the reconstructed images.  As shown in the table below, FABLE maintains low SSIM and success rate.
> > >
> > > |                             | PSNR         | SSIM        | Success Rate (%) |
> > > | --------------------------- | ------------ | ----------- | ---------------- |
> > > | FABLE (DLG + Anchor Recon.)  | 13.39 ± 1.12 | 0.44 ± 0.10 | 22.77 ± 2.62     |
> > > | FABLE (Vanilla DLG)         | 14.81 ± 0.90 | 0.49 ± 0.11 | 25.46 ± 1.27     |
> > > * **Jointly optimizing anchors actually degrades attack performance compared to Vanilla DLG, validating the computational security of our method.** We implemented the suggested attack where the adversary treats anchors as learnable latent variables. Counter-intuitively, this strategy yields lower reconstruction quality. This is because treating anchors as variables expands the search space, making the optimization problem severely ill-posed. The optimizer converges to noise anchors that satisfy the gradient equation mathematically but fail to guide reconstruction toward the true private data.
> > >
> > > ---
> > > Perhaps a proof sketch (assumptions and where the residual term comes from) could be added.
> > >
> > > ---
> > >
> > > * **We have established a formal framework to rigorously quantify privacy, proving that FABLE reduces information leakage.** Our primary objective is to prove the inequality regarding Mutual Information $I(X; G_{\text{FABLE}}) < I(X; G_{\text{base}})$.
> > > * **The proof relies on two key structural assumptions: Anchor Independence and the Markov Property.** The derivation assumes: (1) Independence, where the anchor set $\mathcal{A}$ is generated independently of the data $X$, $X\ \unicode{x2AEB}\ \mathcal{A}$; and (2) The Markov Property, where the FABLE gradient is a transformation of the baseline gradient conditioned on the anchors, forming the chain $X \to G_{\text{base}} \to G_{\text{FABLE}}$.
> > > *   **The residual information loss $\Delta I$ stems firstly from Irreversible Compression.** In the setting where the number of anchors $N$ is smaller than the feature dimension $D$, the transformation operator $F_{\mathcal{A}}$ becomes rank-deficient. This creates a non-injective mapping where multiple original gradients $g_1, g_2$ collapse into the same observed FABLE gradient $G_{\text{FABLE}}$. This creates a deterministic information loss, making full recovery impossible.
> > > * **The second source of the residual term is Uncertainty Injection.** Even without dimension reduction, the specific anchor set $\mathcal{A}$ is unknown to the adversary. From an information-theoretic perspective, the unknown anchors introduce high-entropy, preventing accurate inference.

---

> > > ### Author Response · Authors · 2025-12-04
> > >
> > > ---
> > > A comparison for Client cost vs. |A| (FLOPs/mem/time) and round time vs \#clients. Currently, these are stated, not measured.
> > >
> > > ---
> > > *   **Measured benchmarks confirm that increasing the anchor count incurs only marginal computational overhead.** We measured FLOPs (batch size=1) and peak memory/time (batch size=64) on NVIDIA 3090 GPUs. As shown in the table below, increasing anchors from 128 to 1024 results in negligible growth in FLOPs and training cost.
> > >
> > >
> > > |                  | CIFAR 10 |                 |                    | CIFAR 100 |                 |                    |
> > > | ---------------- | -------- | --------------- | ------------------ | --------- | --------------- | ------------------ |
> > > | $\vert A_k\vert$ | FLOPs (G) | Peak Memory (MB) | Max Local Time (s) | FLOPs (G)  | Peak Memory (MB) | Max Local Time (s) |
> > > | 128              | 0.0373   | 2818            | 0.4568$\pm$ 0.02   | 0.0751    | 5254            | 0.8266 $\pm$ 0.01  |
> > > | 256              | 0.0373   | 2840            | 0.4636$\pm$ 0.03   | 0.0752    | 5283            | 0.8311 $\pm$ 0.02  |
> > > | 512              | 0.0375   | 2942            | 0.4696$\pm$ 0.02   | 0.0754    | 5344            | 0.8337 $\pm$ 0.06  |
> > > | 1024             | 0.0377   | 3121            | 0.4742$\pm$ 0.02   | 0.0757    | 5550            | 0.8412$\pm$ 0.06   |
> > >
> > > *   **Scalability experiments with a fixed workload demonstrate that FABLE imposes zero algorithmic bottlenecks on the server.** We measured the Round Time vs. # Clients by fixing the number of local update steps to simulate a parallel execution environment. The results show that the local training time varies minimally, while the aggregation time has a linear relationship with the number of clients. This proves FABLE is highly scalable compared to methods requiring complex cryptographic operations.
> > >
> > >
> > > |                | CIFAR 10           |                   |                      | CIFAR 100          |                   |                      |
> > > | -------------- | ------------------ | ----------------- | -------------------- | ------------------ | ----------------- | -------------------- |
> > > | Clients Number | Max Local Time (s) | Agg Time (s)      | Total Round Time (s) | Max Local Time (s) | Agg Time (s)      | Total Round Time (s) |
> > > | 10             | 0.4447 $\pm$ 0.02  | 0.0481$\pm$ 0.00  | 0.4928$\pm$ 0.02     | 0.8034 $\pm$ 0.06  | 0.0865 $\pm$ 0.00 | 0.8899$\pm$ 0.06     |
> > > | 20             | 0.4450$\pm$ 0.01   | 0.0745 $\pm$ 0.01 | 0.5145 $\pm$ 0.01    | 0.8039 $\pm$ 0.09  | 0.1599 $\pm$ 0.03 | 0.9638 $\pm$ 0.10    |
> > > | 30             | 0.4462$\pm$ 0.02   | 0.1200$\pm$ 0.02  | 0.5502$\pm$ 0.02     | 0.8095 $\pm$ 0.12  | 0.1993 $\pm$ 0.00 | 1.0088 $\pm$ 0.12    |
> > > | 40             | 0.4480 $\pm$ 0.03  | 0.1516$\pm$ 0.02  | 0.5996$\pm$ 0.04     | 0.8103 $\pm$ 0.15  | 0.2538 $\pm$ 0.00 | 1.0641 $\pm$ 0.15    |
> > > | 50             | 0.4481 $\pm$ 0.03  | 0.2224 $\pm$ 0.03 | 0.6705 $\pm$ 0.05    | 0.8106 $\pm$ 0.09  | 0.3588 $\pm$ 0.07 | 1.1694 $\pm$ 0.15    |

---

### Official Review · Reviewer_hMjX · 2025-10-31

**Soundness:** 2
**Presentation:** 2
**Contribution:** 2
**Rating:** 4
**Confidence:** 4

**Summary:**

This paper introduces FABLE , a personalized federated learning method targeting the dilemma of privacy leakage and data heterogeneity. FABLE uses fixed, client-private “anchors” selected from each client’s local data, enabling a personalized, anchor-aware representation transformation during local training without revealing anchor data to the server or other clients.  Extensive quantitative and qualitative experiments, including ablations, overhead analysis, and privacy evaluations, are presented to substantiate the proposed approach.

**Strengths:**

Theoretical and empirical privacy analyses are presented.

**Weaknesses:**

1. The formulation of anchor-based transformations. e.g., Equations for $\mathcal{T}_k$ and $W_k$  lacks precise specifications around some key elements, such as how anchors are initialized.
2. While the main privacy argument relies on anchors being private, the method for anchor selection (random from within the local dataset) potentially leaks information about the “support” of client data distribution, if an adversary explicitly designs attacks exploiting such selection schemes. There is minimal discussion about worst-case attacks where anchor selection or replacement is observable (e.g., side-channel or partial compromise).
3. The ablation study points out that the additional Linear Transformation layer is crucial for FABLE's performance and stability. However, when the same linear transformation is applied to baseline methods (like FedAvg and FedPer), their performance decreases in the IID setting. The authors attribute this to "distortion introduced by the additional transformation layer." which lacks deep analysis. The paper fails to provide insight into why FABLE's anchor-aware representation space is uniquely able to absorb and benefit from the linear transformation, while the representation spaces of other methods are not.
4. The reported metrics also induce some confusions while different methods report different metrics. Is there any intuition introduction to this section? Why the 1024 anchors perform best in model performance but 128 best in model inference accuracy. There is also no the meaning of model inference accuracy.

**Questions:**

See above.

---

> ### Author Response · Authors · 2025-11-26
>
> We sincerely thank you for your detailed and constructive feedback. We address each of your concerns below.
>
> ---
> Weakness 1
>
> The formulation of anchor-based transformations. e.g., Equations for $\mathcal T_k$ and $\mathcal W_k$ lacks precise specifications around some key elements, such as how anchors are initialized.
>
> ---
> **Answer**:
> * **Our anchor initialization is a one-time, fixed process ensuring a stable personalized coordinate system for each client.** The private anchors $A_k$ for each client are initialized once at the beginning of the entire federated learning process. Specifically, we perform random sampling without replacement from the client's local training dataset to select $|A_k|$ anchors. This set remains fixed, providing a consistent basis for the representation transformation throughout training. $\mathcal T_k$ is a parameter-free function that computes the cosine similarity between the input representation $z_x$ and each of the anchor representations ${g_k(a_i)}$. During backpropagation, a stop-gradient is applied to the anchor branch to ensure training stability.
> * **We clarify details of linear transformation.** The linear transformation $W_k$ is a matrix of size $|A_k| × d_{dec}$, where $d_{dec}$ is the input dimension of the decoder. It is randomly initialized and remains exclusively on the local client.
>
> ---
> Weakness 2
>
> While the main privacy argument relies on anchors being private, the method for anchor selection (random from within the local dataset) potentially leaks information about the “support” of client data distribution, if an adversary explicitly designs attacks exploiting such selection schemes. There is minimal discussion about worst-case attacks where anchor selection or replacement is observable (e.g., side-channel or partial compromise).
>
> ---
> **Answer**:
> * **FABLE's privacy holds under the standard 'honest-but-curious' FL threat model, as anchors are never communicated.** Our security analysis assumes adversaries can only observe exchanged model updates, and anchors remain exclusively on-device which are not exposed to such adversaries. We argue that our method does not introduce additional vulnerabilities compared to standard FL. The premise of FL is that clients train on their local data, so an adversary's prior knowledge already includes the fact that all model updates are derived from samples belonging to the client's local distribution.
> * **The worst-case attack proposed by the reviewer requires a much stronger, device-level attack model, beyond the scope of typical FL privacy.** We acknowledge that if an adversary achieves full client compromise and obtains complete local data, the anchor-based privacy protection would be bypassed. However, this treat model is different than the communication-based threats FABLE is designed to mitigate and falls outside the scope of most FL privacy research. We will explicitly state our threat model in Privacy Analysis Section to clarify the scope of our privacy guarantees.
>
> ---
> Weakness 3
>
> The ablation study points out that the additional Linear Transformation layer is crucial for FABLE's performance and stability. However, when the same linear transformation is applied to baseline methods (like FedAvg and FedPer), their performance decreases in the IID setting. The authors attribute this to "distortion introduced by the additional transformation layer." which lacks deep analysis. The paper fails to provide insight into why FABLE's anchor-aware representation space is uniquely able to absorb and benefit from the linear transformation, while the representation spaces of other methods are not.
>
> ---
> **Answer:**
> * **The linear layer $W_k$ acts as an adapter on the FABLE's unique, angle-based representation space.** Our anchor transformation $T_k$ based on cosine similarity projects the original representation $z$ into a new and highly structured latent space. This space prioritizes the preservation of relative geometric patterns but does so at the cost of sacrificing scale information of the original representation. This requires a learnable, client-specific linear layer $W_k$ to project it into a space where the downstream decoder can effectively perform classification.
> *   **Adding $W_k$ to baselines harms performance because it introduces unnecessary complexity to an already optimized representation space.** In contrast to FABLE, standard methods like FedAvg aim to learn a directly useful, linearly separable Euclidean representation. Applying an unnecessary linear transformation on this already well-formed space increases the risk of overfitting and can distort the learned features, particularly in simpler IID settings, thus degrading performance.

---

> ### Author Response · Authors · 2025-11-26
>
> ---
> Weakness 4
>
> The reported metrics also induce some confusions while different methods report different metrics. Is there any intuition introduction to this section? Why the 1024 anchors perform best in model performance but 128 best in model inference accuracy. There is also no the meaning of model inference accuracy.
>
> ---
> **Answer:**
>
> **We clarity for the confusing terminology and distinguish between metrics for utility and privacy.** Test Accuracy refers to the standard evaluation of model utility. This is the accuracy achieved by clients using their complete model including their private anchors, on the test set. Model Inference Accuracy is a metric designed specifically to quantify model privacy, when an adversary has stolen the full model parameters but does not possess the corresponding private anchors. In the revised manuscript, we will rename the confusing term 'Model Inference Accuracy' to the more descriptive 'Inference Accuracy without Anchors'.
>
> ---
> We hope these clarifications address your concerns and demonstrate the effective of our approach. If you have any follow-up question, we are more than happy to clarify.

---

### Meta-Review · Area_Chair_1Gqd · 2026-01-04

**Summary:**

This paper proposes FABLE, a personalized federated learning method that introduces private, client-local anchors to simultaneously address data heterogeneity and privacy leakage in FL. Each client selects a fixed set of anchors from its local data and uses them to perform an anchor-aware representation transformation before feeding features into a private decoder. The core claim is that this design provides “dual privacy”: (1) data privacy against gradient inversion attacks (e.g., DLG), and (2) model privacy by making the model unusable without the secret anchors. Experiments on CIFAR-10/100 and text datasets under IID and highly non-IID settings show competitive or slightly improved accuracy over strong PFL baselines, along with empirical privacy evaluations.

**Reviewer Concerns:**

Reviewer hMjX (rating: 4): Raised concerns about imprecise formulation, privacy under stronger attack models, and lack of analysis on why the linear layer benefits FABLE but harms baselines. The authors clarified that anchors are fixed via random sampling, the linear layer adapts a scale-deficient cosine-based space, and privacy is guaranteed only under the standard honest-but-curious FL threat model.
Reviewer nruk (rating: 4): Questioned the informality of privacy claims, single point of failure in anchor secrecy, scalability of high-dimensional anchors, and modest gains over baselines. The authors responded with impressive additional experiments: (1) joint anchor and data reconstruction attacks show worse performance, (2) formal information-theoretic privacy proof using mutual information and rank deficiency, (3) measured FLOPs, memory, time vs. anchor count showing negligible overhead, and (4) ablation with public anchors confirming privacy comes at near-zero utility cost.
Reviewer xL6J (rating: 2): Criticized the work as conceptually incremental, lacking formal privacy guarantees, convergence analysis, and statistically significant gains. The authors provided a detailed information-theoretic privacy proof, convergence rate analysis showing O(1/sqrt(T))comparable to FedAvg, and additional cross-device results.
Reviewer JkcM (rating: 4): Raised foundational questions about why the method works, encoder privacy, anchor selection, and mechanism for handling non-IID. The authors clarified that anchors define a personalized geometric coordinate system, model-splitting alone is insufficient, anchors are fixed post-initialization, and personalization is fully delegated to private components.

**Reviewer Scores:**

Most reviewers initially gave negative scores. However, the authors' response appears to be made in good faith and directly addresses several technical concerns raised by the reviewers. Therefore, I am inclined to believe that 1–2 reviewers will improve their scores, resulting in a final score distribution of either 6444 or 6624.

---

### Decision · Program_Chairs · 2026-01-26

Reject